# Time-restricted feeding combined with exercise improves hepatic and glycaemic metabolism in obese mice: A sex-dependent study

Gabriel Calheiros Antunes[1], Camila Venturini Ayres Cunha[2], Leandro Kansuke Oharomari[2], Renan Fudoli Lins Vieira[1], Maura Fanti[3], Thaiane da Silva Rios[2], Luciana Renata Conceição de Mattis[1], Ana Paula Azevêdo Macêdo[1], Adelino Sanchez Ramos da Silva[4], Eduardo Rochete Ropelle[1,5], Dennys Esper Cintra[2,5] and José Rodrigo Pauli[1,5]

[1]*Laboratory of Molecular Biology of Exercise, University of Campinas (UNICAMP), Limeira, São Paulo, Brazil*
[2]*Laboratory of Nutritional Genomics, University of Campinas (UNICAMP), Limeira, São Paulo, Brazil*
[3]*Longevity Institute and Davis School of Gerontology, University of Southern California, Los Angeles, California, USA*
[4]*Postgraduate Program in Rehabilitation and Functional Performance, Ribeirão Preto Medical School, and Postgraduate Program in Physical Education and Sport, School of Physical Education and Sport of Ribeirão Preto, University of São Paulo (USP), Ribeirão Preto, São Paulo, Brazil*
[5]*OCRC – Obesity and Comorbidities Research Center, University of Campinas (UNICAMP), Campinas, São Paulo, Brazil*

*The Journal of Physiology*

Handling Editors: Paul Greenhaff & Max Petersen

The peer review history is available in the Supporting information section of this article (https://doi.org/10.1113/JP287681#support-information-section).

**Abstract figure legend** Metabolic and molecular adaptations of 8 weeks of TRF and TRF+EXE treatments on male and female C57BL/6J fed a western diet.

**Abstract**   Erratic feeding patterns, such as those experienced by shift workers, can exacerbate obesity and metabolic-associated fatty liver disease (MAFLD). Both nutritional factors and sexual dimorphism influence the progression of MAFLD. Time-restricted feeding (TRF) has emerged as a promising strategy to mitigate the effects of obesity, supported by evidence of its

benefits for metabolic disorders like MAFLD. Regular physical exercise is also recommended as a non-pharmacological approach to combat obesity and its related conditions. Both TRF and exercise independently show promise in improving metabolic health, weight management and glycaemic control. Thus, combining these approaches may offer a more effective strategy against obesity and MAFLD. In this study, male and female C57BL/6J mice were subjected to an 8-week obesity induction protocol, followed by TRF (16/8) or TRF combined with aerobic exercise. The results showed that TRF, even during the inactive phase of the mice, had positive effects on weight loss, adiposity, glycaemic homeostasis, insulin sensitivity, liver lipid composition, hepatic fat accumulation and the reduction of lipogenic and inflammatory genes in the liver. The combination of TRF with aerobic exercise provided additional benefits, including improved regulation of hepatic triglycerides and respiratory exchange ratio (RER) in males, enhanced fasting glucose levels in females and reduced *Fatp4* gene expression in both sexes. Aerobic exercise performance also improved in both sexes, with males achieving superior results. Notably, the combination of TRF with aerobic exercise provided greater metabolic benefits, with sex-specific differences observed in metabolic responses.

(Received 12 September 2024; accepted after revision 23 July 2025; first published online 14 August 2025)

**Corresponding author** José R. Pauli: Laboratory of Molecular Biology of Exercise (LaBMEx), School of Applied Science, University of Campinas, 13484–350 – Limeira, SP, Brazil. Email: paulijr@unicamp.br

**Key points**

- Eight weeks of western diet induced obesity, an impaired glucose homeostasis and increased hepatic fat accumulation in male and female C57BL/6J mice.
- Time-restricted feeding (TRF) 16/8 in the active phase and TRF combined with aerobic exercise reduced weight gain and metabolic disorders in C57BL/6J male and female mice fed a western diet.
- TRF when combined with aerobic exercise displayed more pronounced improvements in the hepatic metabolism.
- TRF when combined with aerobic exercise improved liver triglycerides and respiratory exchange ratio in males, fasting glucose in females and decreased lipogenic gene *Fatp4* expression in both males and females.

## Introduction

Metabolic-associated fatty liver disease (MAFLD), characterized by the excessive accumulation of fat in the liver, is the most common liver disease, with an estimated global prevalence of 32% (Eslam et al., 2020; Riazi et al., 2022). Obesity significantly contributes to the development of several diseases, including diabetes, cardiovascular disease and certain types of cancer, thereby increasing the risk of MAFLD (Fazzino et al., 2023; Pi-Sunyer, 2009). Moreover, the progression of MAFLD is influenced by sex due to hormonal differences between men and women (Burra et al., 2021; Lonardo et al., 2019). Despite this, the molecular mechanisms underlying the development of the disease and effective treatment options remain major challenges, as they have not yet been fully elucidated, particularly concerning sex differences.

**Gabriel C. Antunes** is a PhD student in Medical Pathophysiology in the Obesity and Comorbidities Research Center (OCRC) at University of Campinas, Brazil. He has a bachelor's degree in nutrition and completed his master's degree in nutrition, sports and metabolism sciences in the Laboratory of Molecular Biology of Exercise (LabMEx) at University of Campinas. With expertise in molecular biology, he investigates the effects of nutritional approaches and physical exercise on molecular mechanisms for the treatment and prevention of metabolic diseases and for the control of hunger and satiety by neurological mechanisms.

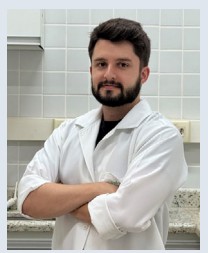

In this context, time-restricted feeding (TRF) has been adopted as a strategy to mitigate the impacts of obesity (Chaix et al., 2019; de Cabo & Mattson, 2019; Wilkinson et al., 2020). Over the past few decades, numerous studies have explored the physiological and molecular effects of this intervention in experimental models, aiming to assess its potential benefits in reducing inflammation and treating metabolic disorders associated with obesity, diabetes and MAFLD (Chaix et al., 2019; Das et al., 2024; Yan et al., 2024). Similarly, physical exercise, as a non-pharmacological approach, has been widely used to minimize or reverse the effects of obesity and its comorbidities (Muñoz et al., 2018; Stine et al., 2023; Yang et al., 2022). Therefore, both TRF and regular physical exercise, when applied independently, appear to offer metabolic benefits, including reductions in body weight and adiposity, as well as improvements in glycaemic homeostasis (Fanti et al., 2021; Oliveira et al., 2023; Vieira et al., 2021).

Although evidence suggests that both TRF and physical exercise independently have positive effects on metabolic health, the combined impact of TRF and physical exercise has been less extensively explored. Furthermore, the effect of TRF on body adiposity and metabolism is influenced by the timing of food intake and by sexual dimorphism (Chaix et al., 2021; Fonken et al., 2010; Salgado-Delgado et al., 2010). Previous studies have shown that restricting access to a high-fat diet during the dark cycle (the active phase of TRF), when rodents typically eat, prevents weight gain and fat accumulation compared with rodents with unrestricted access to high-fat food (Fonken et al., 2010; Salgado-Delgado et al., 2010). While the potential of TRF in avoiding or reducing weight in rodents fed a high-fat diet is well-recognized, Chaix and colleagues demonstrated that the effects on body adiposity may differ between sexes (Chaix et al., 2021). Both young and old male mice were found to have a lower body weight during the dark phase of TRF compared with their *ad libitum*-fed peers, a difference not observed in female mice. This sexual dimorphism in body mass response is not limited to TRF alone; female mice are more resistant to weight gain and the development of insulin resistance when exposed to a high-fat diet compared with males (Maric et al., 2022).

We recently demonstrated that combining TRF (16/8) with aerobic exercise, where male mice had access to food at night – during their most active period – resulted in significant benefits, including reduced body weight gain, decreased fat accumulation in the liver, and improved glycaemic homeostasis (16). Conversely, when TRF was applied during the diurnal cycle, which corresponds to the mouse's least active period, the beneficial effects were either less pronounced or not observed at all (Aoyama & Shibata, 2020; de Goede P, Foppen E, Ritsema WIGR, Korpel NL, Yi CX, 2019; Freire et al., 2020; Hatori et al., 2012).

It is therefore important to investigate whether combining TRF with aerobic exercise can also be effective when applied during the light cycle (i.e. offering food during the day), which corresponds to the animal's less active phase. This inquiry is particularly relevant given that many humans work on an inverted schedule, being more active at night and less active during the day, yet consuming food primarily during daylight hours (Boege et al., 2021; Marjot et al., 2023). Additionally, it is worth noting that in the study by Vieira et al. (2021), aerobic exercise was performed at the end of the active period after the male rodents had finished eating (Vieira et al., 2021). Previous research has shown that eating in the morning or midday followed by evening exercise can more effectively prevent weight gain compared with exercising in the morning followed by eating at noon or in the evening (Pendergrast et al., 2023). Equally important, as previously mentioned, TRF is less effective in female rodents compared with males (Chaix et al., 2021), suggesting that additional interventions, such as an exercise training programme, may be necessary to achieve health improvements and weight loss in female mice fed a high-fat diet. These findings highlight the need for further exploration into the effects of TRF combined with physical exercise, particularly regarding sex-specific outcomes.

We hypothesize that TRF, when combined with aerobic exercise, will produce complementary and synergistic effects, leading to greater improvements in metabolic health than TRF alone, with a corresponding positive impact on body weight, inflammation and metabolism in both male and female mice. Additionally, we speculate that these benefits will be observed even when food access is restricted to the animals' diurnal cycle. To test this, we investigated the effects of TRF combined with aerobic exercise on obesity and liver metabolism in male and female mice fed a western diet during the light cycle. Our findings contribute to the understanding of the combined effects of TRF and aerobic exercise, highlighting sex-specific responses and their implications for obesity and liver metabolism in mice on a western diet.

## Methods

### Ethical approval

All animal protocols were approved by the Animal Ethics Committee (CEUA) of the Institute of Biological Sciences, UNICAMP, Campinas-SP (Protocol number 5957-1/2022), and were conducted following the guidelines of the National Council for Animal Experimentation Control (CONCEA). The study adhered to the ethical principles and reporting standards of *The Journal of Physiology*.

## Experimental protocol

C57BL/6J male and female mice, aged 6 weeks, were obtained from the Multidisciplinary Centre for Biological Investigation on Laboratory Animal Science (CEMIB) at the University of Campinas (UNICAMP). The mice were randomly assigned to four experimental groups for 16 weeks: control mice fed a standard rodent commercial diet *ad libitum* (CTL) ($n = 6$); obese mice fed a western diet *ad libitum* (OB) ($n = 6$); TRF mice fed a western diet and subjected to TRF ($n = 6$); and TRF+EXE mice fed a western diet and subjected to TRF combined with an aerobic exercise protocol ($n = 6$). The number of mice per group was six and equal for both sexes, totalling 48 mice. The diet-induced obesity (DIO) period lasted for the first 8 weeks, during which the mice were fed a western diet, followed by an 8-week obesity treatment phase. The mice were kept on a 12 h light/12 h dark cycle and housed in polyethylene cages with free access to water. The standard diet was a commercial pellet provided by Nuvilab (Quimtia, Colombo, PR, Brazil), with a nutritional composition of 23% crude protein, 4% lipids, 68% carbohydrate and 5% fibre. The western diet consisted of a high-fat diet combined with a high-sucrose solution for hydration. The high-fat diet macronutrient distribution is 20% protein, 35% lipids and 40% carbohydrates, composed of 11.55% corn starch, 20% casein, 10% sucrose, 13.2% dextrinized starch, 4% soybean oil, 31.2% lard, 5% cellulose, 3.5% mineral mix, 1% vitamin mix, 0.3% l-cystine and 0.25% choline bitartrate (Cintra et al., 2012), based on the American Institute of Nutrition (AIN-93G) guidelines (Sundaram & Yan, 2016). The high-sucrose solution was diluted at 42 g/l and consisted of 55% fructose and 45% D-glucose (Synth) (Asgharpour et al., 2016). Throughout the experiment, the animals were weighed weekly, and food intake was measured once a week by assessing the food weight over 24 h.

## Time-restricted feeding protocol

Zeitgeber time (ZT) 0 was designated as lights-on (6 a.m.) and ZT12 as lights-off (6 p.m.). Mice in the TRF and TRF+EXE groups followed a TRF protocol. They had access to food 2 h after lights-on (ZT2 – 8 a.m.) until 2 h before the start of the dark cycle (ZT10 – 4 p.m.), resulting in an 8 h feeding window (Hatori et al., 2012). Food access was managed by transferring the mice between cages daily with free access to a western diet and cages with free access to water only. This protocol allowed the mice 8 h of food access followed by 16 h of food restriction each day during the week, with *ad libitum* food access on the weekends (Fig. 1).

## Incremental load test

After a 5-day adaptation period to aerobic exercise on a treadmill at 3 m/min, mice in the TRF+EXE group underwent an incremental load test. They began running at 6 m/min on a flat treadmill, with speed increasing by 3 m/min every 3 min until exhaustion, defined as touching the end of the treadmill five times within 60 s. The exhaustion velocity determined from this test was used to set the intensity for a subsequent 8-week chronic aerobic exercise protocol at 60% of the exhaustion velocity. The incremental load test was repeated at the end of the 4th and 8th weeks to adjust the running speed as performance improved. Mice were given 24 h of rest between each training session. The effects of aerobic exercise were assessed using a body weight-dependent method (body mass × exhaustion velocity), with performance specifically measured by exhaustion velocity. No adverse effects were observed in any of the mice during the incremental load test (Fig. 2).

## Aerobic exercise training protocol

The aerobic exercise training lasted for 8 weeks, with an intensity set at 60% of the exhaustion velocity obtained from the incremental load test. Mice underwent aerobic exercise for 5 days each week (Monday to Friday) after the feeding period, with rest on weekends. The training sessions were conducted from ZT10 (4 p.m.) to ZT11 (5 p.m.). The exercise duration gradually increased, starting with 30 min in the first week, 45 min in the second week, and 60 min from the third week onward. No adverse outcomes were observed in any of the mice during the incremental aerobic exercise protocol.

## Body weight and Lee Index

Mice were weighed twice a week on an analytical balance scale (L3102I, BEL) to monitor body weight progression. Weight gain was determined using the following formula: Weight gain (g): (Final weight(g) – Initial weight(g)). The Lee Index was carried out through the cubic root of body weight divided by the naso-anal length of the mice ($\sqrt[3]{\text{Weight(g)}}/\text{Length(mm)}$) (dos Santos et al., 2019; Rogers & Webb, 1980).

## Oxygen consumption, carbon dioxide production and respiratory exchange ratio

Between the 6th and 7th weeks of the experiment, mice were subjected to the Comprehensive Lab Animal Monitoring System (CLAMS-Oxymax) (Columbus Instruments, Columbus, OH, USA). Mice were acclimated

to the equipment for 24 h in individual cages and then followed by 24 h of analysis. On the day of the experimental analysis, the TRF+EXE group trained from ZT11 to ZT12, with data collection beginning at ZT12. During the experiment, the mice were housed in separate cages under controlled temperature conditions. Oxygen consumption ($VO_2$), carbon dioxide production ($VCO_2$), respiratory exchange ratio (RER) and spontaneous activity were evaluated (Sasaki et al., 2014). Data were collected six times per hour throughout the 24 h and were subsequently plotted in Microsoft Excel 2016, with data grouped by hour.

## Intraperitoneal glucose and insulin tolerance tests (ipGTT and ipITT)

After 8 h of fasting and 24 h after the last aerobic exercise session, blood samples were collected from the mice's tails to measure basal glucose levels (timepoint zero). A 25% glucose solution (2 g/kg of body weight) was then administered intraperitoneally (IP), with blood samples collected at 30, 60 and 120 min for glucose measurement (Accu-Check Active, Roche, Switzerland). For the intraperitoneal insulin tolerance test (ipITT), recombinant human insulin (Humulin R, Eli Lilly, Indianapolis, IN,

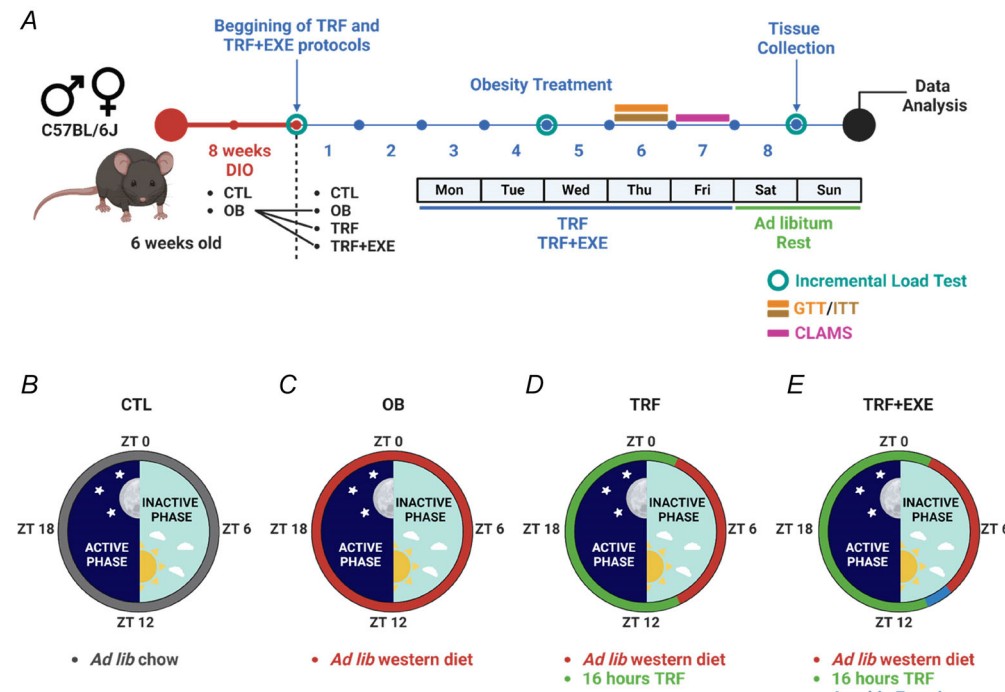

**Figure 1. Experimental design**
*A*, timeline of the protocol: During the first 8 weeks of the experiment, mice were divided into two groups – CTL (mice fed an *ad libitum* chow diet) and OB (mice fed a western diet *ad libitum*) for obesity induction. In the 9th week, the OB group was further divided into three groups: OB (continuing the western diet *ad libitum*), time-restricted feeding (TRF), and time-restricted feeding combined with aerobic exercise (TRF+EXE) for obesity treatment, which continued until tissue collection in the 16th week. Representations of the CTL group (*B*), OB group (*C*), TRF group (*D*) and TRF+EXE group (*E*) are shown.

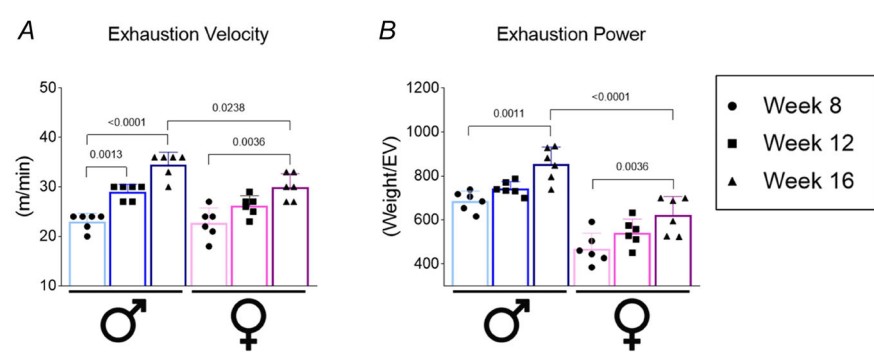

**Figure 2. Aerobic exercise training improved performance in male and female mice**
*A*, exhaustion velocity across the three tests performed during the aerobic exercise protocol. *B*, exhaustion power across the three tests performed during the aerobic exercise protocol. Bars represent the means ± SD for the 1st, 2nd and 3rd tests (*n* = 6).

USA) was administered at a concentration of 1.5 U/kg of body weight via intraperitoneal injection. Blood samples were collected at 0, 10, 15, 20, 25 and 30 min for glucose measurement (Accu-Check Active, Roche, Switzerland). Blood samples were obtained using a tail snip (via surgical scissors), with bleeding controlled between time points using a compression bandage (Johnson & Johnson, São Paulo, SP, Brazil). After the tests, the mice were monitored for up to 3 h.

## Western blotting analysis

Twenty-four hours after the last aerobic exercise session and following a 4 h fast, the mice were given an intra-peritoneal injection of ketamine chlorohydrate (90 mg/kg; Ketalar, Parke-Davis, Ann Arbor, MI, USA) and xylazine (10 mg/kg; Rompun, Bayer, Leverkusen, Germany). Afterward, the mice were decapitated, and the liver was collected and rapidly frozen in liquid nitrogen for storage at −80°C. The tissue was subsequently homogenized in extraction buffer (1% Triton X-100, 100 mM Tris (pH 7.4), 100 mM sodium pyrophosphate, 100 mM sodium fluoride, 10 mM EDTA, 10 mM sodium vanadate, 2 mM phenylmethylsulfonyl fluoride and 0.1 mg/ml aprotinin) at 4°C using a Bead Ruptor 12 Homo-genizer (OMNI International, Kennesaw, GA, USA) operated at maximum speed for 60 s. The lysates were centrifuged in an Eppendorf 5804R (Hamburg, Germany) at 12,851 $g$ at 4°C for 15 min to remove insoluble material. The supernatant was used for the assay, and protein content was determined using the bicinchoninic acid method. Laemmli buffer containing 100 mM dithiothreitol was then added to the super-natant, and the samples were heated for 5–10 min. Next, equal amounts of protein (40 μg) were applied to a poly-acrylamide gel for separation by SDS-PAGE and trans-ferred to nitrocellulose membranes. Ponceau staining was used to verify membrane transfer. The blots were blocked with 5% dry milk at room temperature for 1 h and then incubated overnight at 4°C with the primary antibodies: FAS (#3189S) and $\beta$-actin (#3700S) from Cell Signalling Technology (Danvers, MA, USA). The membranes were then incubated for 1 h with the appropriate secondary antibodies. The specific bands were visualized using enhanced chemiluminescence and quantified by their areas using optical densitometry with UN-SCAN-IT gel 6.1 software (Silk Scientific, Inc., Orem, UT, USA).

## RNA extraction and RT-qPCR

Liver tissue was homogenized in 400 μl Trizol (Thermo Fisher Scientific) and RNA content was extracted according to the manufacturer's instructions. A total of 2 μg of RNA was used for the cDNA synthesis using High-Capacity cDNA Reverse Transcription Kits (Thermo Fisher Scientific). The cDNA samples were subjected to a quantitative real-time polymerase chain reaction (RT-qPCR) using 300 ng cDNA and 0.3 μm primers for *Fasn*, *Srebp1c*, *Cd36*, *Fatp4*, *Nfkb*, *Tnfα*, *Tlr4* and *Il1β*, with *Gapdh* as the endogenous control (Table 1) synthesized by Exxtend (Paulínia, SP, Brazil), and iTaq Universal SYBR Green Supermix (Bio-Rad, Hercules, CA, USA). The data were evaluated using StepOne Software (Thermo Fisher Scientific), calculating $\Delta\Delta Ct$.

## Blood biochemistry analysis

Biochemical markers, including total cholesterol and triglycerides, were analysed in the serum and liver of mice following tissue collection after decapitation (24 h after the last aerobic exercise session and 4 h of fasting) using commercial kits from Laborlab (São Paulo, SP, Brazil). Non-esterified fatty acids (NEFA) were measured in the serum using the Wako HR series NEFA-HR assay from Fujifilm (Richmond, VA, USA).

## Histological analysis

The liver tissue was extracted and cryopreserved in pre-cooled isopentane at −80°C. Thin slices (10 μm) of the liver were cut using a Leica cryostat (CM1850, Heidelberg, Germany) and mounted on adhesion slides. The slides were stained with Oil Red O solution (Sigma-Aldrich, St Louis, MO, USA) for 30 min and haematoxylin for 1 min, then washed and sealed with a gelatin–glycerin solution. Images were captured using Leica Application Suite software. The MAFLD Activity Score was evaluated in a double-blind manner to assess diet-induced liver damage. Steatosis was graded from 0 to 3 based on the percentage of affected tissue, hepatocellular ballooning from 0 to 2 based on the number of ballooning hepatocytes, and lobular inflammation from 0 to 3 based on the number of inflammatory foci. All these parameters contributed to a total MAFLD Activity Score (Kleiner et al., 2005; Power Guerra et al., 2022), with a steatogenic profile evaluated at 400× magnification. Inflammation was characterized by the presence of at least five inflammatory cells not arranged in a row across 10 fields (Liebig et al., 2018).

## Mass spectrometry analysis for fatty acid determination

The chromatographic analysis was conducted using a gas chromatograph coupled with a mass spectrometer (GCMS-QP2010 Ultra; Shimadzu), equipped with an automatic injector (AOC-20i). The chromatographic column was made of fused silica Rt-2560, measuring 100 m in length, 0.25 mm in inner diameter and 0.20 μm

**Table 1. Primer sequences used in the RT-qPCR technique.**

| Gene | Forward | Reverse |
|---|---|---|
| *Fasn* | 5′-GAGGACACTCAAGTGGCTGA-3′ | 5′-GTGAGGTTGCTGTCGTCTGT-3′ |
| *Srebp1c* | 5′-GAGGACACTCAAGTGGCTGA- 3′ | 5′-GGGAAGTCACTGTCTTGGTTGTT-3′ |
| *Cd36* | 5′-TGGAGCTGTTATTGGTGCAG-3′ | 5′-TGGGTTTTGCACATCAAAGA-3′ |
| *Fatp4* | 5′-GACTTCTCCAGCCGTTTCCACA-3′ | 5′-CAAAGGACAGGATGCGGCTATTG-3′ |
| *Nfkb* | 5′-GATTCCGGGCAGTGACG-3′ | 5′-GATGAGGGGAAACAGATCGTCC-3′ |
| *Tnfα* | 5′-CAGGCGGTGCCTATGTCTC-3′ | 5′-CGATCACCCCGAAGTTCAGTAG-3′ |
| *Tlr4* | 5′-GTTCTCTCATGGCCTCCACT-3′ | 5′-GGAACTACCTCTATGCAGGGAT-3′ |
| *Il1β* | 5′-TGGACCTTCCAGGATGAGGACA-3′ | 5′-GTTCATCTCGGAGCCTGTAGTG-3′ |
| *Gapdh* | 5′-AACTTTGGCATTGTGGAAGG-3′ | 5′-ACACATTGGGGGTAGGAACA-3′ |

in thickness, from the brand Restek. Ultrapure helium gas was used as the carrier gas at a flow rate of 1.4 ml/min. A sample of 1 µl was injected at a split ratio of 1:20 (SPLIT). The injector temperature was maintained at 215°C, while the oven heating programme started at 80°C and held for 5 min, with a programmed heating ramp up of 5°C/min until reaching 175°C, followed by a heating rate of 1°C/min until 215°C, where the temperature was held for 26 min. The mass spectrometer operated under the following conditions: ionization voltage of 70 eV, ionization source temperature of 215°C, full scan mode with a range of 35–500 m/z, and a scan speed of 0.2 s per scan.

### Statistical analysis

Data are presented as means $\pm$ standard deviation (SD) and statistical significance was determined as $P < 0.05$. Normality was assessed using the Shapiro–Wilk W-test. For data with a normal distribution, one-way Analysis of Variance (ANOVA) was performed, and for comparisons across different time points, two-way ANOVA was used, followed by Tukey's test. For non-parametric data, the Kruskal–Wallis test was employed. All statistical analyses and graphing were conducted using GraphPad Prism 8.0.

### Results

#### Aerobic exercise training improves performance in both male and female mice

Physical performance was evaluated using the incremental load test conducted in the 8th, 12th and 16th weeks in the TRF+EXE groups. The data showed that both male and female mice experienced a significant increase in exhaustion velocity and power in the final test compared with the initial test, demonstrating that aerobic exercise effectively improved aerobic performance in both sexes. Additionally, a significant difference was observed

between male and female mice in the later tests (Fig. 2*A* and *B*).

#### TRF and TRF combined with aerobic exercise reduce body weight and adiposity in males and females DIO mice

After 8 weeks of DIO, the mice were subjected to 8 weeks of TRF or TRF+EXE. The results showed that DIO effectively increased body weight and weight gain in mice fed a western diet compared with the control group (CTL). Additionally, significant differences in weight were observed in the TRF and TRF+EXE groups but only in male mice (Fig. 3*A*, *B*, *G*, and *H*). Cumulative food intake was higher in the OB group compared with the CTL group, but both TRF and TRF+EXE reduced food intake relative to the OB group in both male and female mice. Furthermore, TRF+EXE decreased food intake compared with the TRF group, but only in male mice (Fig. 3*C*, *I*). Supporting these findings, the Lee Index was elevated in the OB group compared with CTL. TRF and TRF+EXE effectively reduced this parameter compared with the OB group, regardless of sex (Fig. 3*D* and *J*).

Regarding body fat, the OB group showed an increase in visceral and total fat content compared with the CTL group. However, both TRF and TRF+EXE reduced visceral fat tissue in both male and female mice. Notably, TRF and TRF+EXE reduced total fat in female mice, while in male mice, only TRF+EXE was effective (Fig. 3*E*, *F*, *K* and *L*).

#### TRF and TRF combined with aerobic exercise improve glucose homeostasis and lipid markers in the serum of male and female mice, while insulin sensitivity only in male mice

In the 6th week of obesity treatment, glucose tolerance tests (GTT) and insulin tolerance tests (ITT) were performed to assess glucose homeostasis. In the GTT,

the OB group showed a significant increase in the glycaemic curve after glucose injection compared with the CTL group. Both TRF and TRF+EXE were effective in maintaining a lower glycaemic curve than the OB group in both male and female mice (Fig. 4*A* and *H*).

The area under the curve (AUC) for the GTT confirmed these results (Fig. 4*B* and *I*). Additionally, fasting glucose levels were higher in the OB group compared with the CTL group, but the TRF and TRF+EXE groups were protected from these metabolic disturbances in male mice.

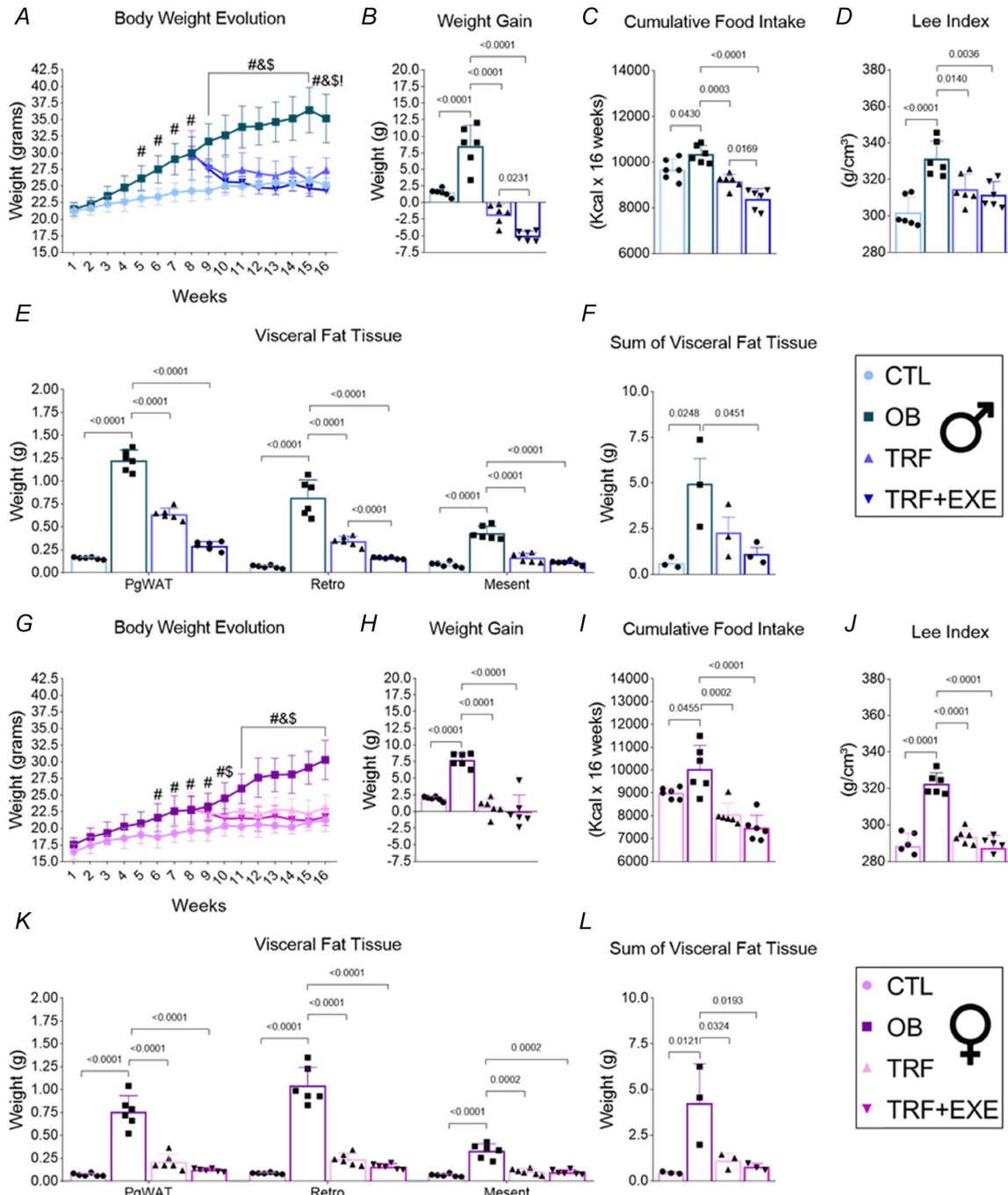

**Figure 3. Time-restricted feeding (TRF) and TRF combined with aerobic exercise reduce weight gain and improve body composition in male and female diet-induced obesity mice**
*A* and *G*, body weight evolution (*n* = 6). *B* and *H*, weight gain (*n* = 6). *C* and *I*, cumulative food intake (*n* = 6). *D* and *J*, Lee Index (*n* = 6). *E* and *K*, visceral fat tissue (*n* = 6). *F* and *L*, sum of visceral fat tissue (*n* = 3). Bars represent the means ± SD. #CTL *vs*. OB; &OB *vs*. TRF; $OB *vs*. TRF+EXE;!TRF *vs*. TRF+EXE.

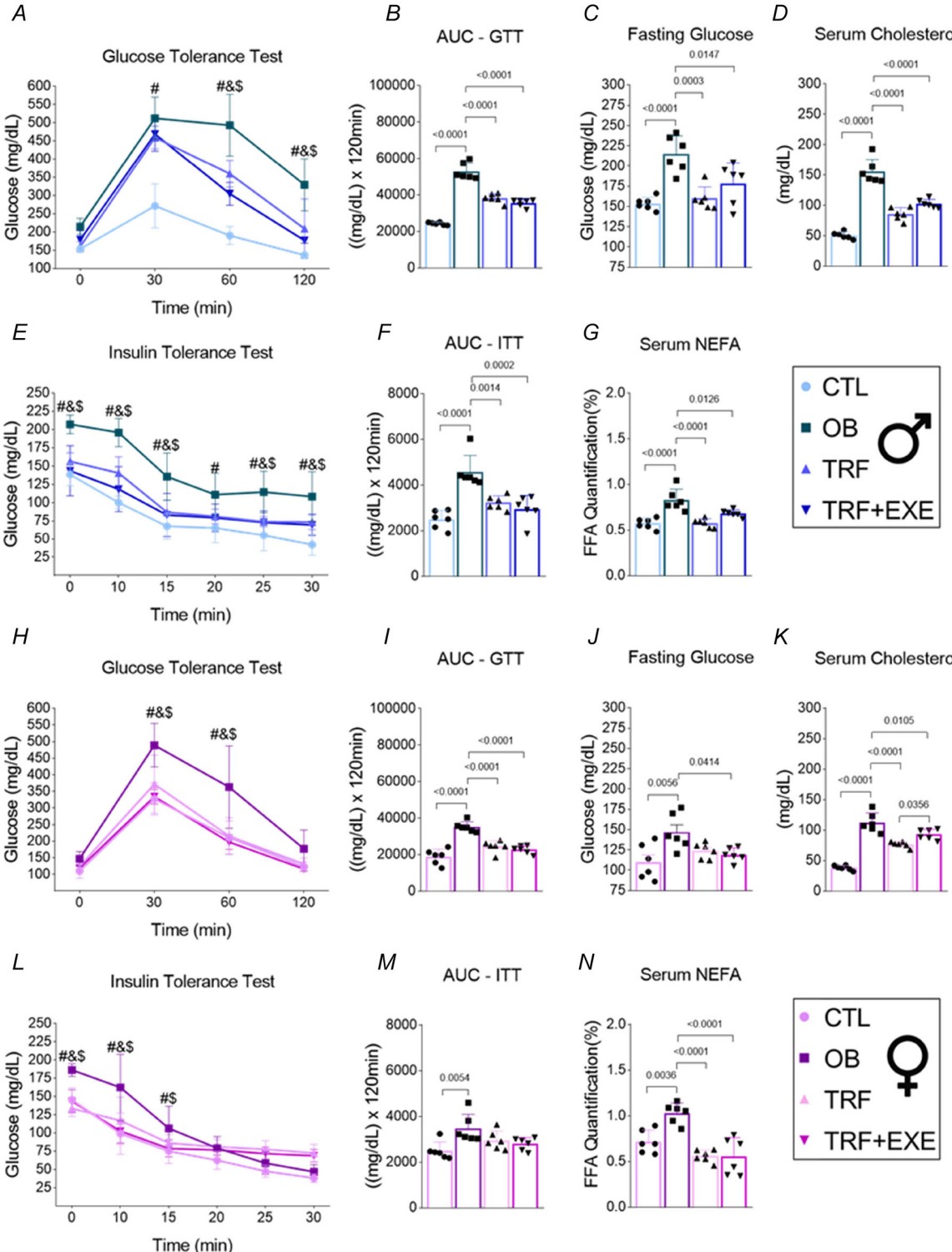

**Figure 4. Effects of time-restricted feeding (TRF) and TRF combined with aerobic exercise on glucose homeostasis and lipid markers in male and female diet-induced obesity mice**

*A* and *H*, glucose tolerance test (GTT) (*n* = 6). *B* and *I*, area under the curve for GTT (*n* = 6). *C* and *J*, fasting glucose (*n* = 6). *D* and *K*, serum cholesterol (*n* = 6). *E* and *L*, insulin tolerance test (ITT) (*n* = 6). *F* and *M*, area under the curve for ITT (*n* = 6). *G* and *N*, serum non-esterified fatty acids (NEFA) (*n* = 6). Bars represent the means ± SD. #CTL *vs*. OB; &OB *vs*. TRF; $OB *vs*. TRF+EXE.

However, in female mice, only TRF+EXE significantly reduced fasting glucose levels (Fig. 4*C* and *J*).

In the ITT, the OB group exhibited a higher glycaemic curve throughout the test in male mice, while in female mice, only the first three time points (0, 10 and 15 min) were higher compared with the CTL group. TRF and TRF+EXE lowered the glycaemic curve compared with the OB group in male mice, but in female mice, TRF and TRF+EXE reduced levels only during the first three time points. The AUC for the ITT supported these findings, showing that TRF and TRF+EXE were effective in lowering values in male mice, even after 8 weeks of DIO. However, in female mice, TRF and TRF+EXE were not effective in reducing the AUC for ITT compared with the OB group (Fig. 4*E*, *F*, *L* and *M*).

After tissue collection, serum was obtained for cholesterol and NEFA analysis. The OB group showed higher levels of cholesterol and NEFA, reflecting the negative effects of DIO in both male and female mice. However, TRF and TRF+EXE were effective in reducing these levels compared with the OB group in both sexes (Fig. 4*D*, *G*, *K* and *N*). These findings indicate that 8 weeks of DIO effectively impaired glucose homeostasis and caused metabolic dysfunctions in lipid metabolism in mice fed a western diet. Nevertheless, TRF, both alone and in combination with aerobic exercise, proved to be an effective strategy for restoring glucose homeostasis and lipid metabolism in male and female mice. However, the restoration of insulin sensitivity was achieved only in male mice.

## TRF and TRF combined with aerobic exercise reduce hepatic fat accumulation and improve lipid metabolism in both male and female mice

For histological analysis, a significant fat accumulation was identified in the OB group compared with the CTL group in both male and female mice. In contrast, TRF and TRF+EXE led to a significant reduction in hepatic fat accumulation compared with the OB group in both sexes (Fig. 5*A* and *E*). These findings align with the MAFLD score, liver cholesterol and triglycerides (TG), which were elevated in the OB group as compared with the CTL group. However, TRF and TRF+EXE effectively reduced the MAFLD score, liver cholesterol and TG levels relative to the OB group in both sexes. Additionally, TRF+EXE was more effective than TRF alone in reducing liver TG levels, but this effect was observed only in male mice (Fig. 5*B*, *C*, *D*, *F*, *G* and *H*). Additionally, we compared the content of triacylglycerols (TG) and liver cholesterol in male and female mice (Supplementary Fig. 1SA and 1SB). The results revealed that CTL and OB female mice showed higher levels of liver cholesterol than CTL and OB

male mice. For liver TG, the CTL and OB groups had no differences between the sexes (Fig. 1*A* and *B*).

## TRF and TRF combined with aerobic exercise alter lipogenic and inflammation genes and lipogenic protein expression in the liver of male and female mice

At the molecular level, the expression of lipogenic genes (*Fasn*, *Srebp1c*, *Cd36* and *Fatp4*) and inflammation-related genes (*Nfkb*, *Tnfα*, *Tlr4* and *Il1β*) was analysed. *Fasn* expression was elevated in the OB group compared with the CTL group in both sexes, but TRF and TRF+EXE effectively reduced their expression only in male mice. Similarly, *Srebp1c* expression was higher in the OB group of both sexes, and TRF and TRF+EXE significantly reduced this lipogenic marker in both male and female mice. *Cd36* expression was markedly increased in the OB group compared with CTL, but TRF and TRF+EXE reduced its expression only in male mice. Regarding *Fatp4* expression, no differences were observed between the OB and CTL groups in either sex. However, TRF and TRF+EXE effectively reduced *Fatp4* expression in male mice compared with the OB group, but not in females. Additionally, TRF+EXE proved more effective than TRF alone in reducing *Fatp4* expression in male mice. Interestingly, TRF alone increased *Fatp4* expression in female mice compared with the OB group, but this effect was mitigated in the TRF+EXE group (Fig. 6*A* and *G*).

For the inflammation markers, *Nfkb* expression increased in the OB group compared with CTL, but only in female mice. Conversely, *Nfkb* was downregulated in the TRF and TRF+EXE groups compared with OB, but exclusively in male mice. Both OB groups exhibited higher levels of *Tnfα* compared with the control, but TRF and TRF+EXE reduced *Tnfα* levels only in male mice. *Tlr4* expression was elevated in the livers of OB male mice, and TRF effectively reduced its expression in both male and female mice. However, TRF+EXE reduced *Tlr4* expression only in male mice. *Il1β* expression was higher in OB male mice compared with CTL, but TRF effectively reduced *Il1β* levels in both sexes. In contrast, TRF+EXE was effective in lowering *Il1β* expression only in male mice compared with the OB group (Fig. 6*D* and *J*).

To confirm these findings, we performed a western blot analysis to evaluate the protein expression of the lipogenic marker FAS. The results showed that FAS content was elevated in the OB group in both sexes compared with the CTL group. TRF was effective in reducing FAS content in both male and female mice. However, TRF+EXE was a particularly potent strategy for reducing FAS content, but only in male mice (Fig. 6*B*, *C*, *E*, *F*, *H*, *I*, *K* and *L*).

These data align with the histological findings, where male mice exhibited greater fat reduction in the TRF and

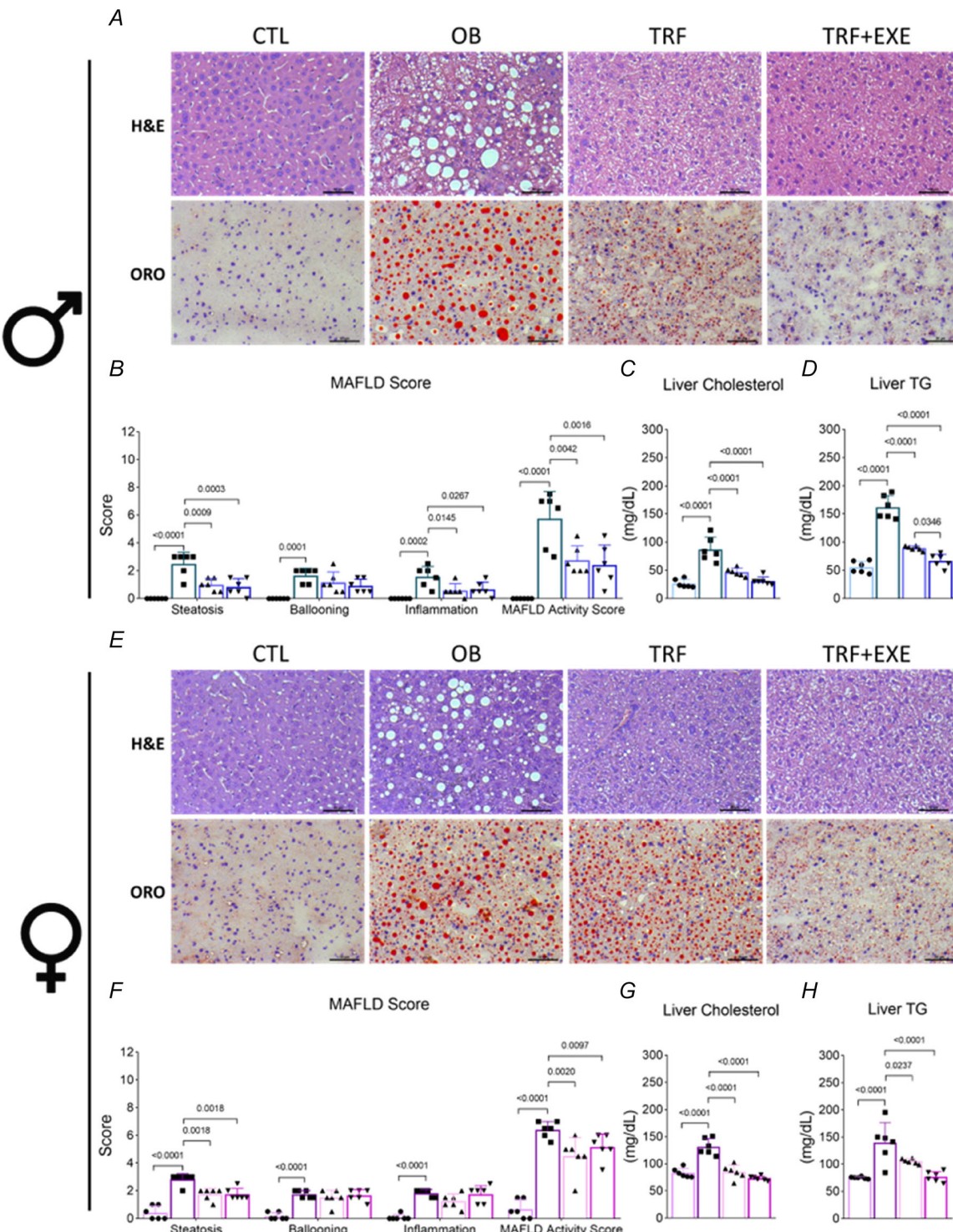

**Figure 5. Time-restricted feeding (TRF) and TRF combined with aerobic exercise improve hepatic fat accumulation in male and female diet-induced obesity mice**

*A* and *E*, H&E and Oil Red O (ORO) staining of the liver. *B* and *F*, MAFLD score (*n* = 6). *C* and *G*, liver cholesterol (*n* = 6). *D* and *H*, liver triglycerides (*n* = 6). Bars represent the means ± SD.

TRF+EXE groups compared with female mice. This effect may be attributed to a more pronounced reduction in lipogenic and inflammation gene expression, as well as lipogenic protein levels, in male mice following TRF and TRF combined with aerobic exercise interventions. These data suggest that the sexes exhibit different metabolic adaptations to these interventions.

## TRF combined with aerobic exercise improves RER only in male mice

Mice were placed in metabolic cages to evaluate $VO_2$, $CO_2$ production and the RER. No differences were observed in the $VO_2$ measurements for either male or female mice (Fig. 7A–D). In the $CO_2$ analysis, a reduction in $CO_2$

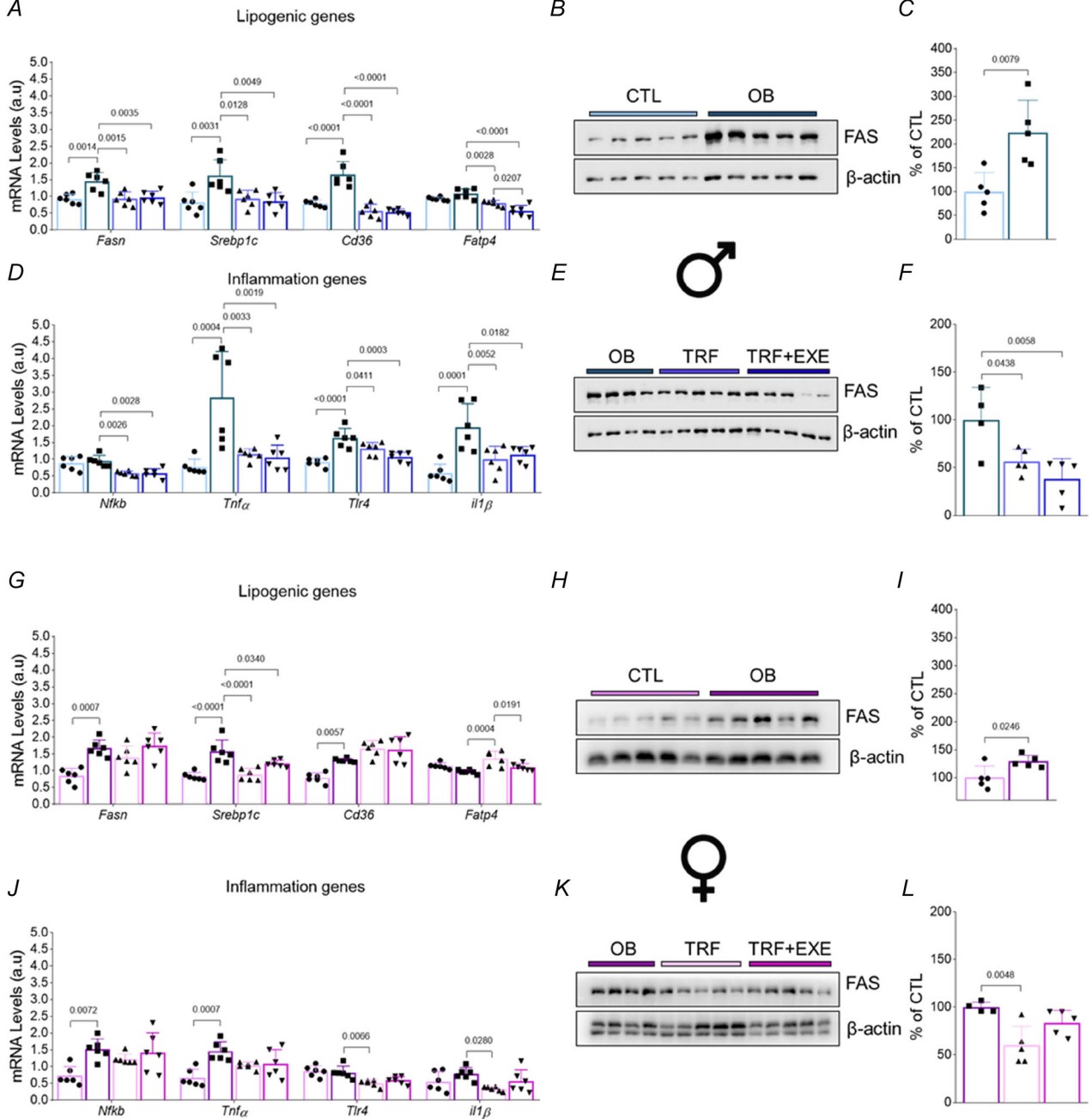

**Figure 6. Time-restricted feeding (TRF) and TRF combined with aerobic exercise improve hepatic lipid metabolism and inflammation of male and female diet-induced obesity mice**
A, G, lipogenic gene expression in liver tissue ($n = 6$). B, C, E, F, H, I, K, L, total FAS protein content in liver tissue for CTL ($n = 5$) *vs.* OB ($n = 5$) and OB ($n = 4$) *vs.* TRF ($n = 5$) *vs.* TRF+EXE ($n = 5$). D, J, inflammation gene expression in liver tissue ($n = 6$). Bars represent the means ±SD.

production was observed in the OB group compared with CTL during both cycles, but an increase was noted in the TRF+EXE group compared with the OB group during the light cycle in male mice (Fig. 7*E* and *F*). In female mice, a reduction in $CO_2$ production was observed only in the TRF group compared with OB (Fig. 7*G* and *H*).

Regarding RER, there was a reduction in both cycles in the OB group compared with CTL for both male and female mice. TRF and TRF+EXE increased RER compared with the OB group during the light cycle in male mice, while in females, this increase was observed only in the TRF+EXE group. Additionally, TRF+EXE reduced RER compared with the TRF group in male mice. During

the dark cycle, RER was decreased in the TRF+EXE group compared with OB and TRF in male mice, while in females, both TRF and TRF+EXE effectively reduced RER compared with the OB group (Fig. 7*I–L*). Regarding the results related to spontaneous activity, it was found that only female mice from the TRF and TRF+EXE groups showed increased activity during the light cycle compared with female mice from the OB group. In the dark cycle, male mice in the TRF+EXE group showed reduced spontaneous activity compared with male mice in the OB group. Concerning female mice, the results revealed that in the dark cycle, only mice in the TRF group

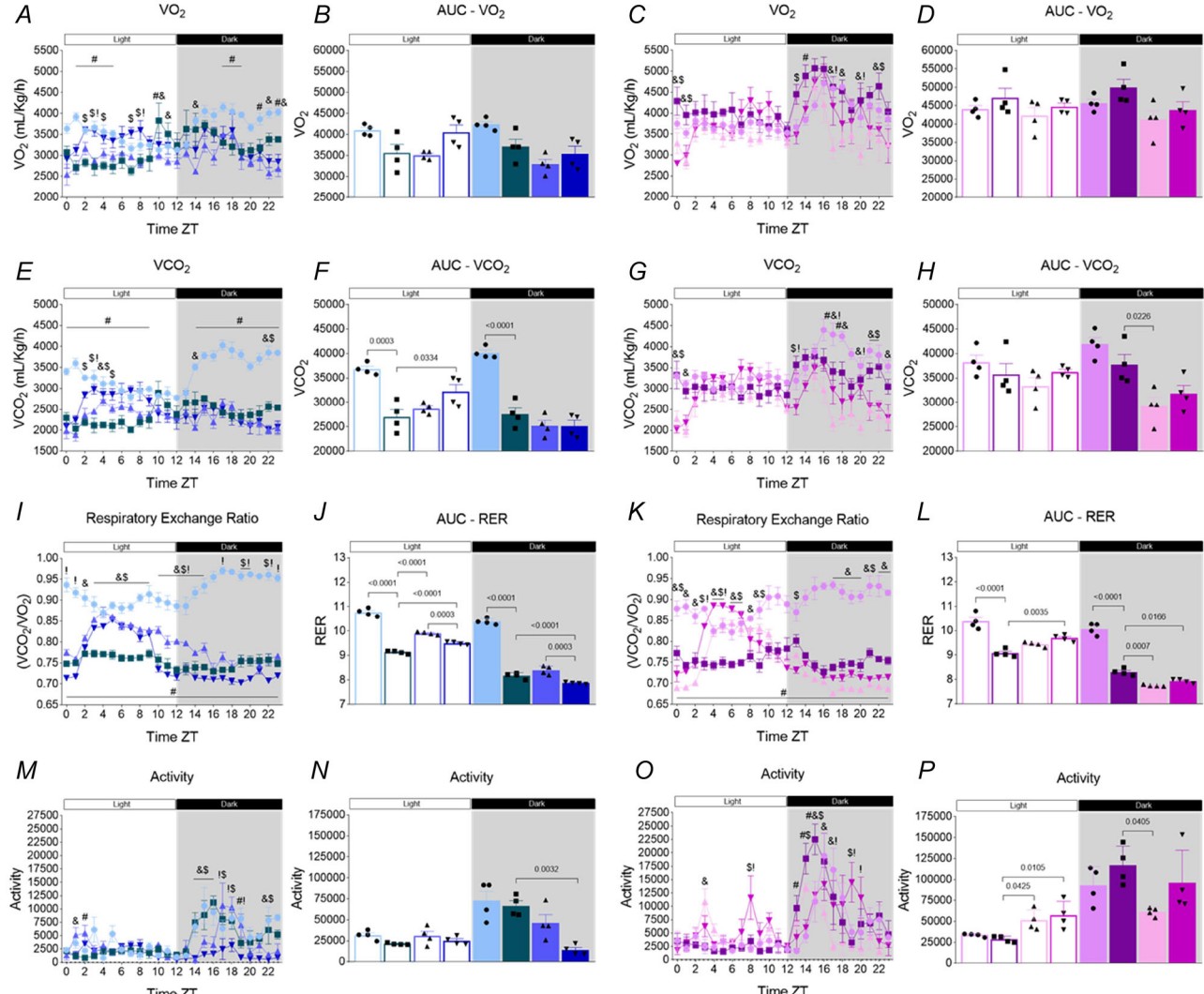

**Figure 7. Effects of time-restricted feeding (TRF) and TRF combined with aerobic exercise on oxygen consumption, carbon dioxide production, respiratory exchange ratio and spontaneous activity**
*A* and *C*, $VO_2$ consumption over 24 h (*n* = 4). *B* and *D*, area under the curve (AUC) of $VO_2$ during the light and dark cycles (*n* = 4). *E* and *G*, $VCO_2$ production over 24 h (*n* = 4). *F* and *H*, AUC of $VCO_2$ during the light and dark cycles (*n* = 4). *I* and *K*, respiratory exchange ratio (RER) over 24 h (*n* = 4). *J* and *L*, AUC of RER during the light and dark cycles (*n* = 4). *M* and *O*, activity over 24 h (*n* = 4). *N* and *P*, AUC of activity during the light and dark cycles (*n* = 4). Bars represent the means ± SD. #CTL *vs*. OB; &OB *vs*. TRF; $OB *vs*. TRF+EXE; !TRF *vs*. TRF+EXE.

showed lower spontaneous activity than mice in the OB group.

## TRF and TRF combined with aerobic exercise alter the fatty acid profile in the liver of male and female mice

Chromatography–mass spectrometry was performed on the liver of male and female mice to evaluate the fatty-acid profile. Inflammatory saturated fatty acids did not increase in the OB group compared with CTL; however, only TRF was able to reduce their levels in male mice. In females, no differences were identified between groups. Consequently, the female TRF group exhibited higher levels than the male TRF group.

C12:0 showed no differences between groups in both males and females, but when comparing sexes, the female TRF group had higher values than the male TRF group. C14:0 and C15:0 were elevated in OB compared with CTL; however, only TRF+EXE effectively reduced C14:0 levels compared with OB. Regarding C15:0, both TRF and TRF+EXE were effective in reducing its levels compared with OB in male mice. In female mice, OB exhibited higher C14:0 levels than CTL, with a reduction observed only in TRF+EXE, while no differences were detected in C15:0

across groups. Thus, the female TRF group appeared to have higher levels compared with the male TRF group (Fig. 8A–D).

For monounsaturated fatty acids, no significant differences were identified between groups. However, the female TRF group exhibited levels higher than the male TRF group. C16:1 n7, C16:1 n9 and C18:1 n9 were elevated in the OB group compared with CTL, but TRF and TRF+EXE reduced their levels compared with OB in male mice. In females, C16:1 n9 and C18:1 n9 were higher in OB compared with CTL. TRF+EXE showed a reduction compared with OB in both C16:1 n9 and C16:1 n7. Overall, C16:1 n7, C16:1 n9 and C18:1 n9 were higher in the female TRF group compared with the male TRF group (Fig. 8E–H).

Polyunsaturated fatty acids were reduced in TRF compared with OB in male mice, with no differences observed in females. Consequently, the female TRF group had higher values than the male TRF group. C20:2 n6 and C22:5 n6 were elevated in OB compared with CTL, while no differences were detected in C20:3 n9. However, TRF+EXE reduced C20:2 n6 levels compared with OB, and TRF reduced C22:5 n6 levels compared with OB. In females, C20:2 n6, C20:3 n9 and C22:5 n6 were elevated in OB compared with CTL. TRF+EXE lowered C20:3

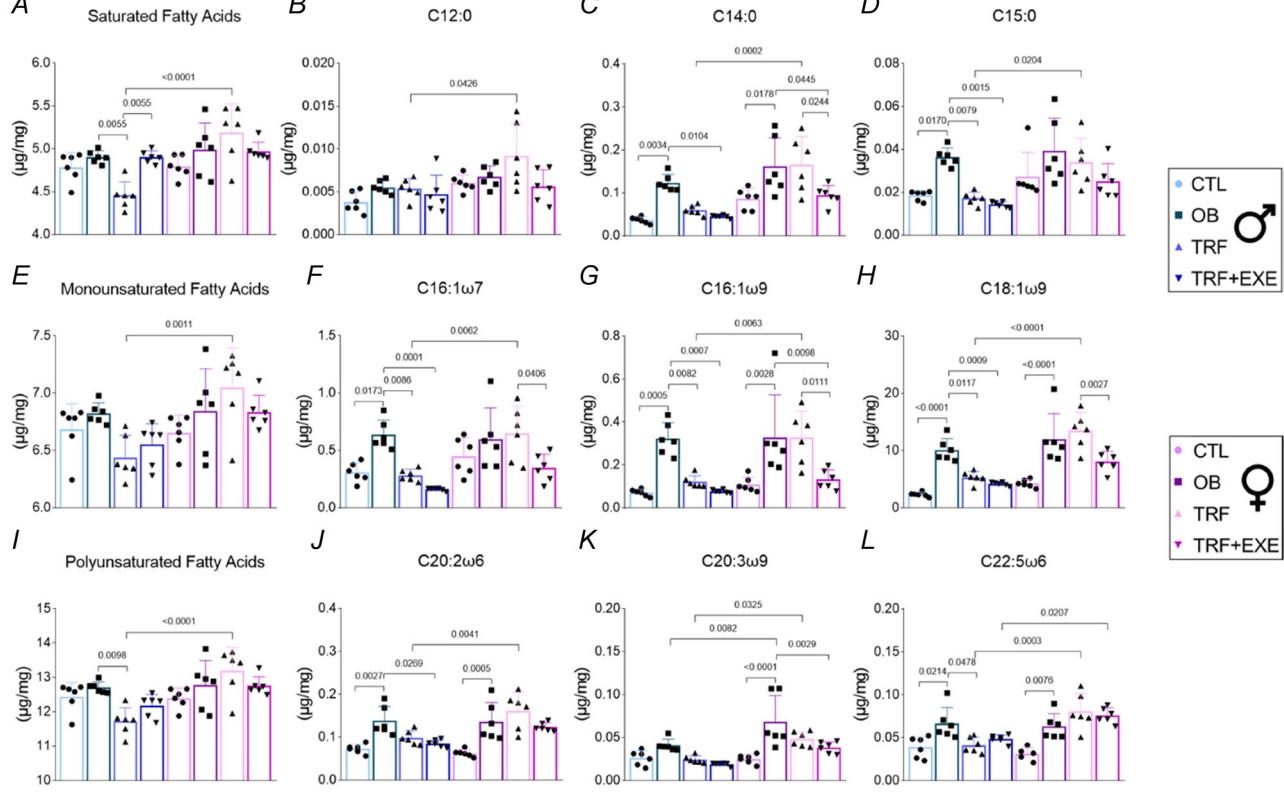

**Figure 8. Effects of time-restricted feeding (TRF) and TRF combined with aerobic exercise on lipid profile**
*A–D*, saturated fatty acids (*n* = 6). *E–H*, monounsaturated fatty acids (*n* = 6). *I–L*, polyunsaturated fatty acids (*n* = 6). Bars represent the means ± SD.

n9 levels compared with OB. Female OB levels were higher than male OB for C20:3 n9, and the female TRF group exhibited higher values than the male TRF group for C20:2 n6, C20:3 n9 and C22:5 n6. Additionally, the female TRF+EXE group had higher levels than the male TRF+EXE group (Fig. 8*I–L*).

### TRF and TRF combined with aerobic exercise alter the ω6:ω3 profile in the liver of male and female mice

The total omega-3 (ω3) profile did not show any differences between groups or sexes. However, the TRF group exhibited a decrease in C20:5 n3 compared with OB. In females, TRF was elevated compared with OB in C22:6 n3, while in C20:5 n3 TRF+EXE was reduced compared with TRF. The female TRF group had higher levels than the male TRF group in both C20:5 n3 and C22:6 n3, while the female TRF+EXE group showed higher levels than the male TRF+EXE group only in C22:6 n3 (Fig. 9*A–C*). The total omega-6 (ω6) profile also did not show any significant differences. However, OB was elevated compared with CTL in C18:2 n6 and C18:3 n6. In male mice, TRF and TRF+EXE reduced C18:2 n6 levels compared with OB, but only TRF+EXE decreased C18:3 n6 levels compared with OB. In females, OB was elevated compared with CTL only in C18:3 n6. TRF increased C18:3 n6 levels compared with OB, while TRF+EXE reduced them. The female TRF group had higher levels than the male TRF group in both C18:2 n6 and C18:3 n6, whereas the female TRF+EXE group showed higher levels than the male TRF+EXE group only in C18:3 n6

(Fig. 9*D–F*). The ω6:ω3 ratio was higher in the male TRF group compared with the female TRF group (Fig. 9*G*).

### Discussion

The present study aimed to investigate the effects of TRF and TRF combined with aerobic exercise during the light phase in male and female mice fed a western diet, focusing on sex-dependent outcomes. The detrimental effects of a diet high in saturated fats and sugars on metabolism, leading to obesity, type 2 diabetes and MAFLD, are well established (Eslam et al., 2020; Fazzino et al., 2023). Additionally, not only the quantity and quality of the diet matter but also the timing of feeding plays a crucial role in metabolism (Guan et al., 2018). Eating at inappropriate times has been shown to exacerbate the negative impacts of obesity in both rodents and humans (Antunes et al., 2010; Arble et al., 2009; Marot et al., 2023). A study by Arble et al. found that C57BL/6J mice fed a high-fat diet during the light phase gained significantly more weight than those fed during the dark phase despite consuming the same amount of calories (Arble et al., 2009). This outcome suggests that feeding at the wrong time may amplify the adverse effects of a high-fat diet, contributing to the development of obesity and its associated diseases. This phenomenon is likely due to circadian disruption, where inappropriate feeding patterns cause a misalignment of the central circadian clock, leading to several negative metabolic outcomes (Freire et al., 2020; Mason et al., 2020). In humans, an increased risk of coronary heart disease has been observed

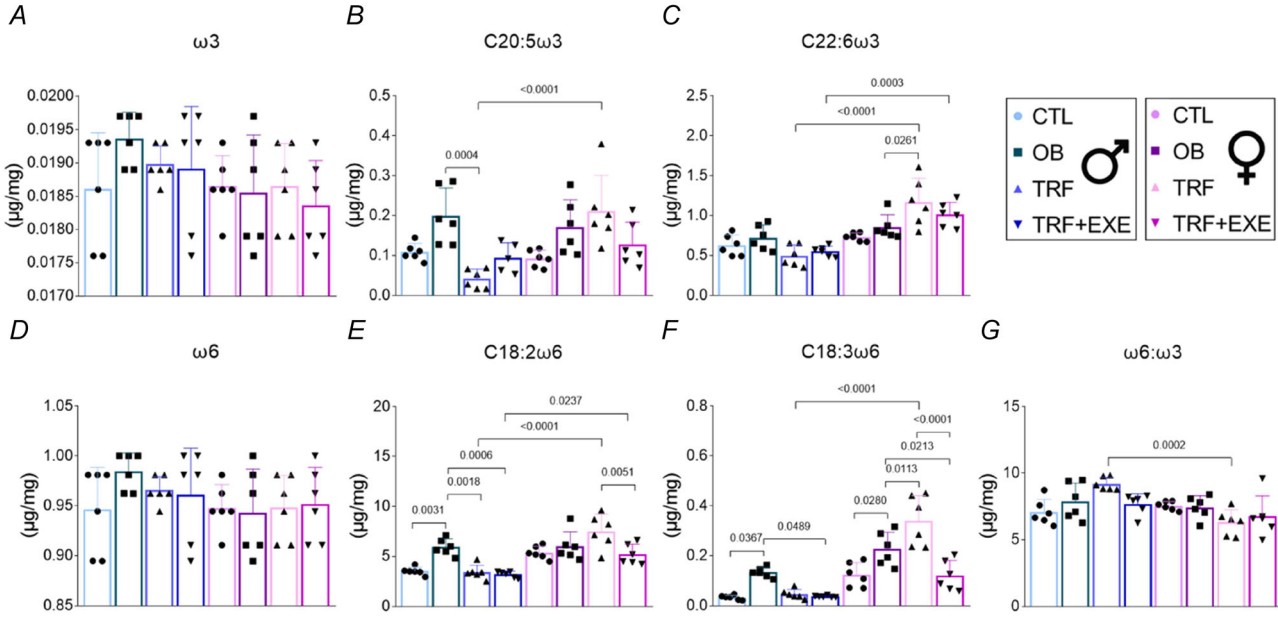

**Figure 9. Effects of time-restricted feeding (TRF) and TRF combined with aerobic exercise on ω6 and ω3 profile**
*A–C*, ω3 fatty acids (*n* = 6). *D–F*, ω6 fatty acids (*n* = 6). *G*, ω6:ω3 ratio (*n* = 6). Bars represent the means ± SD.

in women who work night shifts, further highlighting the impact of circadian misalignment (Patterson & Sears, 2017; Vetter et al., 2016).

Non-pharmacological approaches such as nutritional therapies and physical exercise are promising strategies for regulating feeding times and maintaining caloric balance. Fasting has been shown to effectively reduce cancer and obesity in rodents, while human studies suggest that fasting may help reduce hypertension and obesity (Longo & Panda, 2016). A previous study from our lab demonstrated the significant benefits of combining TRF and aerobic exercise in Swiss male mice. Mice subjected to an 8 h TRF during the dark phase, combined with aerobic exercise, showed improvements in body weight, hepatic fat accumulation, fatty acid oxidation and reduced lipogenesis. Additionally, insulin signalling pathways were restored in the TRF combined with aerobic exercise group compared with TRF alone, and TRF did not compromise aerobic exercise performance (Vieira et al., 2021).

Similarly, in our current study, both male and female mice exhibited improved performance after 8 weeks of aerobic exercise, even when combined with TRF. However, a significant difference was observed in the final sessions between sexes, with males displaying higher exhaustion velocity and power than females. This result aligns with known sex-specific characteristics, as men typically have greater strength, power and speed than women of the same age due to differences in endogenous metabolism (Hunter et al., 2023). A study by Chaix et al. (2021) further highlighted the sex-dependent benefits of a 9 h TRF regimen in C57BL/6J male and female mice. Both sexes were protected from glucose intolerance and hepatic fat accumulation. Still, only male mice showed a reduction in body weight despite similar caloric intake between the *ad libitum* and TRF groups (Chaix et al., 2021).

In our study, similar findings were observed, with mice subjected to 8 h of food availability during the light phase showing a reduction in body weight and weight gain only in male mice, while TRF prevented weight gain in female mice. Freire and collaborators reported similar results, where male mice gained more weight than female mice, although their protocol focused on prevention rather than treatment, as in our study (Freire et al., 2020). Moreover, TRF combined with aerobic exercise had an additional effect in male mice compared with those subjected only to TRF. In our study, we chose to implement aerobic running training shortly after the exposure period to the western diet. This timing may have provided an advantage for performing the proposed physical exercise and may have also contributed to reduced energy storage in the form of fat. This interpretation is supported by the fact that contracting muscles require energy to sustain their activity. In future studies, it will be important to investigate whether exercising in a fasted state poses a challenge to running performance and whether this impacts the effects of combining TRF with physical exercise. Contrary to the findings of Chaix et al. (2019), our TRF and TRF+EXE groups showed a significant reduction in cumulative food intake, regardless of sex (Chaix et al., 2021). Additionally, caloric consumption in the TRF+EXE group was lower than in the TRF group, but only in male mice, which could explain the weight reduction observed in these groups. This caloric difference may be attributed to the fact that the mice were fed during their inactive phase (Freire et al., 2020).

Regarding glucose metabolism and lipid markers, both sexes showed improvements. However, in female mice, only the TRF+EXE group demonstrated an improvement in fasting glucose levels. This result contrasts with a study conducted in Wistar rats, which found improved glucose tolerance only in rats fed during the nighttime (de Goede et al., 2019). On the other hand, another study identified enhanced glucose tolerance in both male and female C57BL/6J mice subjected to a 6 h TRF during the active phase (Freire et al., 2020). However, for insulin sensitivity, only male mice showed improvement. It is important to note that female mice did not show improvement in the ITT test, possibly because they did not develop insulin resistance due to less weight gain than the male mice.

Another consideration is fasting glucose levels. Although the OB group showed a significant increase in fasting glucose compared with the CTL group in female mice, the values were considerably lower than in male mice, suggesting that 8 weeks of DIO may not have been sufficient to induce insulin resistance in female mice. Research indicates that females have more natural protection against the development of insulin resistance than males (Palmisano et al., 2017; Zhu et al., 2013). This protection is partly due to differences in fat distribution; females tend to have more subcutaneous fat, while males accumulate more intra-abdominal fat, which is associated with higher cardiometabolic risk and the development of type 2 diabetes. Additionally, women exhibit greater insulin sensitivity than men, due to its higher content and activation of proteins (IR, Akt and GLUT4) related to glucose uptake in the adipose and muscle tissue (Nicolaisen et al., 2024). Estrogen also plays a crucial role in metabolic homeostasis, and its decline during menopause can reduce this protective effect (Della Torre & Maggi, 2017; Palmisano et al., 2017). This protective effect is also evident in the liver, where the prevalence of MAFLD is higher in men than in women of the same age. However, post-menopausal women lose this estrogen-related protection, placing them at the same risk of MAFLD as men (Burra et al., 2021).

In liver metabolism, TRF was effective in reducing fat accumulation regardless of sex. When combined with aerobic exercise, it was even more effective in reducing hepatic fat accumulation. Male C57BL/6J mice subjected to an intermittent fasting protocol (1 day fed, 1 day

fasting) for 4 weeks after 8 weeks of DIO on a high-fat or high-fructose diet showed a significant reduction in hepatic steatosis and inflammation (Marinho et al., 2019). Our data similarly show a reduction in steatosis and inflammation, as evidenced by the MAFLD score, leading to decreased MAFLD activity in the TRF and TRF+EXE groups, particularly in male mice. In females, while steatosis was reduced in the TRF and TRF+EXE groups, overall MAFLD activity was still decreased compared with the OB group.

Another study from our lab also demonstrated a reduction in fat accumulation, as shown by Oil Red staining and the MAFLD score, in male Swiss mice subjected to TRF and TRF combined with resistance exercise for 8 weeks (de Lima et al., 2023). Together with our data, these findings indicate that TRF is effective in reducing or preventing hepatic fat accumulation, liver triglycerides, cholesterol and serum lipid markers in mice fed a western or high-fat diet. However, when combined with exercise, these markers showed a more pronounced reduction compared with TRF alone (Chaix et al., 2021; de Lima et al., 2023; Vieira et al., 2021). Additionally, these effects may vary depending on sex. A study by Oliveira and collaborators also highlighted the enhanced impact of combining TRF with exercise (Oliveira et al., 2023).

Complementary to our findings, Freire et al. (2020) demonstrated sex-specific adaptations in body composition and hepatic metabolism in C57BL/6J male and female mice subjected to 6 h of food access during the active phase for 4 weeks. Their study revealed a significant difference in fat deposition in the liver, with female mice showing higher triglyceride content than male mice. These data were also identified in our study, where CTL and OB female groups presented higher levels of liver cholesterol than the respective groups of male mice, but no differences in liver TG. In our experiment, feeding a western diet for 8 weeks during the inactive phase led to increased fat deposition in both male and female mice (Freire et al., 2020), indicating that treatment with TRF, and particularly TRF combined with aerobic exercise, was more effective in male mice than in females, as observed in H&E and Oil Red staining. Additionally, male mice showed slightly lower levels of liver cholesterol and triglycerides than female mice. These findings align with the study by Freire et al., which reported enhanced lipogenesis and triglyceride storage in female mice compared with males when food was restricted to 6 h during the active phase (Freire et al., 2020). Furthermore, while TRF+EXE effectively reduced liver triglycerides in both sexes, the complementary effect of aerobic exercise combined with TRF was observed only in male mice.

Consistent with previous findings, we evaluated lipogenic and inflammation markers. Lipogenic genes (*Fasn*, *Srebp1c*, *Cd36*) were elevated in both sexes within the OB group, as confirmed by increased FAS protein

levels compared with controls. Inflammation markers showed that OB male mice had elevated levels of *Tnfα*, *Tlr4* and *Il1β*, while female mice exhibited increases in *Nfkb* and *Tnfα*. Mice fed a western or high-fat diet have been shown to upregulate lipogenic and inflammation markers (Marinho et al., 2019). Additionally, short-term feeding during the inverted cycle (lights on) has been shown to exacerbate obesity and increase lipogenic markers in the liver of mice fed a western diet for 1 week, compared with those fed during the correct time (lights off) (Yasumoto et al., 2016).

Time-restricted feeding, both alone and combined with aerobic exercise, downregulated lipogenic markers in male mice but was less effective in females, where only *Srebp1c* decreased. Interestingly, the TRF+EXE group showed downregulation of *Fatp4* compared with TRF alone, regardless of sex. TRF significantly reduced FAS protein levels in the OB group, with no additional reduction observed in females when combined with aerobic exercise. Inflammation markers also showed reduced expression with TRF and TRF combined with aerobic exercise in male mice, while in females, only *Tlr4* and *Il1β* levels decreased with TRF.

TRF has been demonstrated to be an effective strategy for restoring hepatic lipid metabolism through the downregulation of lipogenic genes like *Fasn* (Chaix & Zarrinpar, 2015). However, the effects of TRF varied by sex. Notably, while lipogenic gene expression was similar between males and females, inflammation gene expression differed significantly. This discrepancy may be attributed to the protective effects of estrogen, which helps shield females from developing hepatic inflammation (Burra et al., 2021; Lonardo et al., 2019). A similar effect was observed in male and female mice fed a high-fructose diet, where both sexes displayed similar hepatic fat accumulation, but inflammation was lower in females (Spruss et al., 2012).

Although no significant differences were observed in the $VO_2$ graph, a statistical reduction in $VCO_2$ was identified in the OB group during both the light and dark phases, along with an increase in $VCO_2$ production in the TRF+EXE group of male mice during the light phase. In female mice, a reduction in $VCO_2$ production was observed in the TRF group only during the dark phase. Regarding the RER, the OB group showed a reduction in this parameter for both males and females in both phases. TRF and TRF+EXE increased RER during the light phase in male mice, while in females, only TRF+EXE led to an increase, which aligns with the feeding time during the light phase (Bray et al., 2013). Additionally, TRF+EXE demonstrated a supplementary effect in reducing RER compared with TRF alone in male mice. During the dark phase, only TRF+EXE reduced RER, with values lower than those in the TRF group in male mice. In female mice, both TRF and TRF+EXE resulted in decreased RER values.

These findings are consistent with previous studies (Gallop et al., 2023; Hatori et al., 2012; Woodie et al., 2018), where fat utilization is more pronounced during the fasting state, corresponding to the dark phase. In female mice, this greater effect was observed with TRF alone, while in males, only the TRF+EXE group showed this result, likely due to adaptations from aerobic exercise, suggesting that TRF has a direct effect on controlling hepatic lipid metabolism and inflammation by regulating genes involved in lipogenic and inflammatory pathways, as well as in the control of cholesterol and triglycerides. Furthermore, when combined with aerobic exercise, TRF may be even more effective. It is important to note that these effects may vary according to sex due to metabolic dimorphism. The analysis of spontaneous activity revealed that, during the light cycle, only female mice from the TRF and TRF+EXE groups exhibited greater activity than those in the obese group. Furthermore, spontaneous activity was consistently higher in the dark cycle than in the light cycle for both males and females. Given that rodents are nocturnal, greater spontaneous activity in the dark cycle is expected for both sexes. However, male mice in the TRF+EXE group were less active during the dark cycle compared with those in the OB and TRF groups, whereas female mice in the TRF+EXE group maintained their activity levels. Our results align with previous findings by Hatori et al. (Hatori et al., 2012), who reported increased spontaneous activity in both male and female mice during the dark cycle.

In general, alterations in the lipid profile were observed only in male mice from the TRF group, specifically for saturated and polyunsaturated fatty acids, while no differences were detected in females. More specifically, the inflammatory fatty acids C14:0 and C15:0 were reduced in the TRF group, and C15:0 was further reduced in the TRF+EXE group in male mice. In females, however, only TRF+EXE was capable of reducing C15:0 levels (Chiappini et al., 2016, 2017). For all saturated fatty acids, the TRF female group exhibited higher levels compared with the TRF male group. Additionally, the anti-inflammatory fatty acids C20:5 n3, C22:6 n3, C18:2 n6 and C18:3 n6 were also higher in female mice than in male mice, indicating a sex-dependent factor (Simopoulos, 2002).

Our study found that an 8-week western diet induced obesity and impaired hepatic metabolism in both male and female mice. However, TRF and TRF combined with aerobic exercise effectively regulated body weight, glucose homeostasis, hepatic fat accumulation, and lipogenic and inflammation markers, thereby alleviating the metabolic damage caused by obesity. Significant metabolic improvements were observed with the combination of TRF and aerobic exercise, including reduced liver triglycerides and RER in males, improved fasting glucose in females, and decreased *Fatp4* expression in both sexes. Notably, these improvements were achieved even though TRF and TRF combined with aerobic exercise were implemented only 5 days per week, with *ad libitum* feeding on weekends. Additionally, aerobic exercise performance improved in both sexes, with males showing superior results. Differences in metabolic responses related to glycaemic homeostasis, such as fasting blood glucose, glucose tolerance tests and hepatic lipid profile, were noted between the sexes. These findings underscore the need for further studies to explore these effects in both male and female subjects.

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

## Additional information

### Data availability statement

The data that support the findings of this study are available from the corresponding author upon reasonable request.

### Competing interests

None declared.

### Author contributions

G.C.A. performed all the experiments. G.C.A. and J.R.P. analysed the data. G.C.A. prepared the figures. G.C.A. and J.R.P. drafted the manuscript. C.V.A.C., L.K.O. and R.F.L.V. participated in the tissue collection, T.S.R. performed the chromatography, M.F., L.R.C.M., A.P.A.M., A.S.R.S., E.R.R., D.E.C. and J.R.P. edited and revised the manuscript. G.C.A. and J.R.P. conceived and designed the research. All authors approved the submitted version.

### Funding

This work was supported by National Council for Scientific and Technological Development (CNPq; case number 309 268/2023-0 and 441 725/2023-6), Coordination for the Improvement of Higher Education Personnel (CAPES; finance code 001) and São Paulo Research Foundation (FAPESP; case numbers 2021/13 847-8; 2019/11 820-5; 2022/0 8930-6; 2024/16 630-8).

### Acknowledgements

The Article Processing Charge for the publication of this research was funded by the Coordenação de Aperfeiçoamento de Pessoal de Nível Superior - Brasil (CAPES) (ROR identifier: 00x0ma614).

### Keywords

aerobic exercise, metabolic-associated fatty liver disease (MAFLD), obesity, sex dimorphism, time-restricted feeding (TRF)

### Supporting information

Additional supporting information can be found online in the Supporting Information section at the end of the HTML view of the article. Supporting information files available:

**Peer Review History**

