## [Peer Review History · The Journal of Physiology]

Time-Restricted Feeding Combined with Exercise Improves Hepatic and Glycemic Metabolism in Obese Mice: A Sex-Dependent Study.

Gabriel Calheiros Antunes, Camila Venturini Ayres Cunha, Leandro Kansuke Oharomari, Renan Fudoli Lins Vieira, Maura Fanti, Thaiane da Silva Rios, Luciana Renata Conceição de Mattis, Ana Paula Azevêdo Macêdo, Adelino Sanchez Ramos da Silva Silva, Eduardo R Ropelle, Dennys Esper Cintra, and Jose Rodrigo Pauli

DOI: 10.1113/JP287681

Corresponding author(s): Jose Pauli (paulijr@unicamp.br)

The following individual(s) involved in review of this submission have agreed to reveal their identity: Ali Javaheri (Referee #2); Ken D O'Halloran (Referee #3)

Review Timeline:

Submission Date:	12-Sep-2024
Editorial Decision:	21-Nov-2024
Revision Received:	13-May-2025
Editorial Decision:	04-Jul-2025
Revision Received:	10-Jul-2025
Editorial Decision:	15-Jul-2025
Revision Received:	15-Jul-2025
Accepted:	23-Jul-2025

Senior Editor: Paul Greenhaff

Reviewing Editor: Max Petersen

Transaction Report:

Dear Dr Pauli,

Re: JP-RP-2024-287681 "Time-Restricted Feeding Combined with Exercise Improves Hepatic and Glycemic Metabolism in Obese Mice: A Sex-Dependent Study." by Gabriel Calheiros Antunes, Camila Venturini Ayres Cunha, Leandro Kansuke Oharomari, Renan Fudoli Lins Vieira, Maura Fanti, Luciana Renata Conceição de Mattis, Ana Paula Azevêdo Macêdo, Adelino Sanchez Ramos da Silva Silva, Eduardo R Ropelle, Dennys Esper Cintra, and José Rodrigo Pauli

Thank you for submitting your manuscript to The Journal of Physiology. It has been assessed by a Reviewing Editor and by 2 expert referees and we are pleased to tell you that it is potentially acceptable for publication following satisfactory major revision.

LANGUAGE EDITING AND SUPPORT FOR PUBLICATION: If you would like help with English language editing, or other article preparation support, Wiley Editing Services offers expert help, including English Language Editing, as well as translation, manuscript formatting, and figure formatting at www.wileyauthors.com/eoo/preparation. You can also find resources for Preparing Your Article for general guidance about writing and preparing your manuscript at www.wileyauthors.com/eoo/prepresources.

REVISION CHECKLIST:

We look forward to receiving your revised submission.

Yours sincerely,

Paul Greenhaff
Senior Editor
The Journal of Physiology

EDITOR COMMENTS

Reviewing Editor:

Methods Details:
See specific requests from referees.

Comments for Authors to ensure the paper complies with the Statistics Policy:
Please replace SEM with SD through and provide precise P values.

Comments to the Author:
Thank you for submitting your research to J Physiol. Your paper has been reviewed by two expert scientists. Strengths included the thorough phenotyping and inclusion of both sexes. Weaknesses included its descriptive nature and a few methodological issues that were raised.

Senior Editor:

Comments to the Author:
Thank you for the manuscript submission to The Journal of Physiology (TJP). It has been considered by a reviewing editor and two expert reviewers. Merit is seen in the work by all in the whole (including deep phenotyping and inclusion of both sexes), but a number of concerns have been raised which the authors should consider, including the addition of data to improve the interpretation and impact of the findings reported.

REFEREE COMMENTS

Referee #1:

The authors present an interesting study examining the effects of TRF and exercise on male and females. While highly descriptive the information that the authors present is still important to the understanding of the dimorphic response of males and females to various stimuli.

There are a few questions and points of clarification that should be considered.

In the methods it is unclear how many males and females were used. In the group descriptions it simply states n=6, but it is unclear if that is 6 of each sex or 6 animals total.

The total % of each macronutrient would be helpful to include with the descriptions of the diets.

The authors might consider including something like the HOMA-IR and fasting insulin levels for the insulin sensitivity this might alleviate some of the variability of the females in the TRF groups.

It would be interesting to see the direct comparison of the male and female at least CTL and OB groups for the liver cholesterol and liver TG. They appear higher which is consistent with other literature.

Did the authors consider the fat composition differences between the sexes. Ramanathan et al 2022 showed that there are differences in liver lipid composition in males and females on a HFD undergoing TRF.

Was activity altered in the mice undergoing TRF or TRF+ exercise. It would be interesting to know if activity patterns were increased or shifted in the mice in the different treatment groups particularly since they were being fed during the light cycle. This would also give insight into any shift in circadian function.

Referee #2:

Summary:

The authors have performed a thorough and interesting evaluation of metabolic shift after time restricted fasting and aerobic exercise in mice. They have found time-restricted feeding and time-restricted feeding accompanied by exercise regulates glucose homeostasis with different results in males and females.

Comments to improve the paper:

1. The authors' study protocol allows mice to access food over weekends, which may counteract the metabolic adaptations induced by time-restricted feeding. Given the rapid metabolic response to altered feeding patterns, maintaining consistent time-restricted feeding throughout the study period would better isolate the effects of this intervention.
2. The study protocol implements exercise following the feeding period, which may produce different metabolic response compared to exercise performed in a fasted state. Since the timing of exercise in relation to feeding can alter metabolic responses (as they have briefly mentioned this in the introduction section), it would be helpful for the authors to justify this choice or include a rationale for how post-feeding exercise was expected to affect outcomes in the methods as well.
3. Providing the exact macronutrient composition (percentage of fat, protein, and carbohydrate) for both normal chow and high-fat/Western diet would improve reproducibility and clarity. Including an ordering number for these diets would further enhance transparency.
4. The rationale behind combining a high-fat diet with a high-sucrose solution for hydration needs clarification. Sucrose can bypass glycolytic regulation (phosphofructokinase), potentially confounding glucose homeostasis results. Including a control group with a non-sucrose hydration option would help isolate the impact of the high-fat diet.

END OF COMMENTS

May 09, 2025

Dear Kim E. Barrett,

Editor-in-chief of The Journal of Physiology

Firstly, we would like to thank you for the opportunity to revise our paper. We have read the reviewer's comments and judged the suggestions very appropriately. We are confident that the edits within this revision have not only satisfied the reviewer's concerns but have significantly improved the manuscript. We hope that our efforts are evident from the responses below and the revised manuscript. All changes in the current document are highlighted in yellow. Also, we have provided a point-by-point commentary for the reviewer's comments. We hope our efforts to reach the conditions you have criticized would be enough to have the paper accepted for publication at The Journal Physiology.

Sincerely, José Rodrigo Pauli, PhD.

REVIEWERS' COMMENTS:

Reviewing Editor:

Methods Details:

See specific requests from referees.

We are thankful for this commentary and all requests from the referees were attended.

Comments for Authors to ensure the paper complies with the Statistics Policy:

Please replace SEM with SD through and provide precise P values.

We agreed and we changed all statistics according to the journal specifications.

In the new version of the manuscript, we included Thaiane da Silva Rios, who was responsible for carrying out all lipid profile analyses.

Referee #1:

The authors present an interesting study examining the effects of TRF and exercise on male and females. While highly descriptive the information that the authors present is still important to the understanding of the dimorphic response of males and females to various stimuli.

There are a few questions and points of clarification that should be considered.

1. In the methods it is unclear how many males and females were used. In the group descriptions it simply states n=6, but it is unclear if that is 6 of each sex or 6 animals total.

We are grateful for this commentary and agree with the suggestion. We added this information to the manuscript. The number of mice per group was 6 and equal for both sexes, totaling 48 mice (Page 4, lines 127-128).

2. The total % of each macronutrient would be helpful to include with the descriptions of the diets.

We agree and we added this information in the manuscript (Page 4, lines 133-135). The information was updated and the macronutrient distribution of each macronutrient from the diet is available separated from the sucrose solution as described by Asgharpour and collaborators, 2016.

The standard diet was a commercial pellet provided by Nuvilab (Quimtia, Colombo, PR, Brazil), with a nutritional composition of 23% crude protein, 4% lipids, 68% carbohydrate, and 5% fiber. The Western diet consisted of a high-fat diet combined with a high-sucrose solution for hydration. The high-fat diet macronutrient distribution is 20% protein, 35% lipids, and 40% carbohydrates, composed of 11.55% corn starch, 20% casein, 10% sucrose, 13.2% dextrinized starch, 4% soybean oil, 31.2% lard, 5% cellulose, 3.5% mineral mix, 1% vitamin mix, 0.3% l-cystine, and 0.25% choline bitartrate (Cintra et al., 2012), based on the American Institute of Nutrition (AIN-93G) guidelines (Sundaram & Yan, 2016). The high-sucrose solution was diluted at 42 g per liter and consisted of 55% fructose and 45% D-glucose (Synth®) (Asgharpour et al., 2016) (Page 4, lines 140).

3. The authors might consider including something like the HOMA-IR and fasting insulin levels for the insulin sensitivity this might alleviate some of the variability of the females in the TRF groups.

We are thankful for this commentary and appreciate the consideration. We understand the need and the importance of those experiments. However, the samples (serum) were insufficient or compromised due to their utilization for previous analysis. Therefore, it was not possible to perform serum insulin analysis.

4. It would be interesting to see the direct comparison of the male and female at least CTL and OB groups for the liver cholesterol and liver TG. They appear higher which is consistent with other literature.

We are grateful for this commentary and agree with the suggestion. We added this information in the supplementary material and mentioned it in the manuscript (Page 15, lines 372-376 and Page 26, lines 631 - 633).

5. Did the authors consider the fat composition differences between the sexes. Ramanathan et al 2022 showed that there are differences in liver lipid composition in males and females on a HFD undergoing TRF.

We appreciate this comment and your consideration. Ramanathan's results are interesting and add important data to the literature. In our study, we performed mass spectrometry coupled with gas chromatography and evaluated the lipid composition of the liver of male and female mice. We believe that these data specifically from mice in the TRF and TRF+EXE groups, both males and females, are new in the literature and can add a lot to future studies. These results were added to the manuscript (Figure 8 and Figure 9, Page 20 – 23, lines 458 – 518 and Page 28, lines 701 - 709). As can be seen, there is a sexual dimorphism in the lipid profile in the liver

6. Was activity altered in the mice undergoing TRF or TRF+ exercise. It would be interesting to know if activity patterns were increased or shifted in the mice in the different treatment groups particularly since they were being fed during the light cycle. This would also give insight into any shift in circadian function.

We are grateful for this commentary and agree with the suggestion. For this, we verified the spontaneous activity in mice submitted to the interventions and added this information to the manuscript (Page 19 -20, line 441 - 447).

Referee #2:

The authors have performed a thorough and interesting evaluation of metabolic shift after time restricted fasting and aerobic exercise in mice. They have found time-restricted feeding and time-restricted feeding accompanied by exercise regulates glucose homeostasis with different results in males and females.

Comments to improve the paper:

1. The authors' study protocol allows mice to access food over weekends, which may counteract the metabolic adaptations induced by time-restricted feeding. Given the rapid metabolic response to altered feeding patterns, maintaining consistent time-restricted feeding throughout the study period would better isolate the effects of this intervention.

We appreciate all your consideration and professional assessment. Without a doubt, the metabolic adaptations of time-restricted feeding could be identified as a chronic and uninterrupted application. However, the main idea of the project was to investigate whether there is a benefit from TRF even when carried out during the rodent's light cycle and with ad libitum feeding during the weekend, to simulate the daily routine of shift workers with weekends off. But we appreciate the reviewer's note and will consider it in future experiments.

2. The study protocol implements exercise following the feeding period, which may produce different metabolic response compared to exercise performed in a fasted state. Since the timing of exercise in relation to feeding can alter metabolic responses (as they have briefly mentioned this in the introduction section), it would be helpful for the authors to justify this choice or include a rationale for how post-feeding exercise was expected to affect outcomes in the methods as well.

Firstly, we would like to thank the reviewer for his consideration. This is an intriguing subject that deserves special attention. In our study, we chose to perform physical exercise immediately after the rodents' feeding period. The main reason for this was to ensure that the animals had the energy to perform the physical exercise protocol. As can be seen from the results, the mice showed an increase in running performance. Another factor we considered was that aerobic training carried out shortly after the end of the period of exposure to the diet (western diet) may have contributed to lower energy storage in the form of fat. This association can be explained because the contracting muscle needs energy to maintain its activity. However, again, the analysis of the impacts of TRF associated with physical exercise performed while fasting needs to be explored. Finally, as suggested by the reviewer, we included the justification for choosing this intervention model with an exercise protocol carried out after the feeding period in the new version of the manuscript

3. Providing the exact macronutrient composition (percentage of fat, protein, and carbohydrate) for both normal chow and high-fat/Western diet would improve reproducibility and clarity. Including an ordering number for these diets would further enhance transparency.

We agree and we added this information in the manuscript (Page 4, lines 133-135). The information was updated and the macronutrient distribution of each macronutrient from the diet is available separated from the sucrose solution as described by Asgharpour and collaborators, 2016.

The standard diet was a commercial pellet provided by Nuvilab (Quimtia, Colombo, PR, Brazil), with a nutritional composition of 23% crude protein, 4% lipids, 68% carbohydrate, and 5% fiber. The Western diet consisted of a high-fat diet combined with a high-sucrose solution for hydration. The high-fat diet macronutrient distribution is 20% protein, 35% lipids, and 40% carbohydrates, composed of 11.55% corn starch, 20% casein, 10% sucrose, 13.2% dextrinized starch, 4% soybean oil, 31.2% lard, 5% cellulose, 3.5% mineral mix, 1% vitamin mix, 0.3% l-cystine, and 0.25% choline bitartrate (Cintra et al., 2012), based on the American Institute of Nutrition (AIN-93G) guidelines (Sundaram & Yan, 2016). The high-sucrose solution was diluted at 42 g per liter and consisted of 55% fructose and 45% D-glucose (Synth®) (Asgharpour et al., 2016) (Page 4, lines 140).

4. The rationale behind combining a high-fat diet with a high-sucrose solution for hydration needs clarification. Sucrose can bypass glycolytic regulation (phosphofructokinase), potentially confounding glucose homeostasis results. Including a control group with a non-sucrose hydration option would help isolate the impact of the high-fat diet.

We are grateful for this commentary. However, the main idea was to evaluate the Western diet effect not the carbohydrate solution by itself. We understand the mechanism driven by the glycolytic pathway, but it was not the aim of the study to explore this pathway. Also, as we used the study of Chaix et al., 2021 as a model, the Western diet was chosen to keep the protocol more similar. For that, as we are on a high-fat diet, the combination

of a sucrose solution is a feasible option to create a Western diet (Asgharpour, 2016). In this way, we decided to not repeat the experiment and keep the results as a Western diet-dependent effect. In any case, the reviewer's suggestion will be considered in future experiments.

Dear Dr Pauli,

Re: JP-RP-2025-287681R1 "Time-Restricted Feeding Combined with Exercise Improves Hepatic and Glycemic Metabolism in Obese Mice: A Sex-Dependent Study." by Gabriel Calheiros Antunes, Camila Venturini Ayres Cunha, Leandro Kansuke Oharomari, Renan Fudoli Lins Vieira, Maura Fanti, Thaiane da Silva Rios, Luciana Renata Conceição de Mattis, Ana Paula Azevêdo Macêdo, Adelino Sanchez Ramos da Silva Silva, Eduardo R Ropelle, Dennys Esper Cintra, and Jose Rodrigo Pauli

Thank you for submitting your manuscript to The Journal of Physiology. It has been assessed by a Reviewing Editor and by 2 expert referees and we are pleased to tell you that it is acceptable for publication following satisfactory revision.

REVISION CHECKLIST:

Please upload two versions of your manuscript text: one with all relevant changes highlighted and one clean version with no changes tracked. The manuscript file should include all tables and figure legends, but each figure/graph should be uploaded as separate, high-resolution files. The journal is now integrated with Wiley's Image Checking service. For further details, see: <https://www.wiley.com/en-us/network/publishing/research-publishing/trending-stories/upholding-image-integrity-wileys->

image-screening-service

We look forward to receiving your revised submission.

Yours sincerely,

Paul Greenhaff
Senior Editor
The Journal of Physiology

EDITOR COMMENTS

Reviewing Editor:

Thank you for choosing the Journal of Physiology for your research. You have addressed the reviewer comments in a satisfactory way. Congratulations on this important contribution to the growing literature on metabolic effects of time-restricted feeding.

Senior Editor:

Thank you for addressing the comments raised in the earlier review. The reviewers and reviewing editor are of the opinion that the manuscript is acceptable for publication. Please note however to comply with The Journal of Physiology statistics policy the authors must amend the manuscript, including the Figures to show exact p values (not *, **, *** in the Figures). Thank you for considering The Journal of Physiology to publish your work.

REFEREE COMMENTS

Referee #1:

The authors present an interesting investigation of the sexually dimorphic responses to time restricted feeding in mice. Although the content is largely descriptive the recent revisions have enhanced the quality and the impact of the study making it a valuable contribution to the growing body of literature in this area.

Referee #3:

Thank you for submitting your manuscript to The Journal of Physiology. The manuscript was selected for ethics review to confirm appropriate use of anaesthetic agents. The authors have provided details of the agents used, the doses used and the route of administration. The ketamine/xylazine mix is standard and appropriate. The authors did not perform surgical procedures. Rather, the mice were immediately euthanised by decapitation. The text is adequate as is. Thank you.

Line 136: Please add that the study conformed to the principles of The Journal of Physiology.

END OF COMMENTS

July 07, 2025

Dear Kim E. Barrett,

Editor-in-chief of The Journal of Physiology

Firstly, we would like to thank you for the opportunity to revise our paper. We have read the reviewer's comments and judged the suggestions very appropriately. We are confident that the edits within this revision have not only satisfied the reviewer's concerns but have significantly improved the manuscript. We hope that our efforts are evident from the responses below and the revised manuscript. We would like to point out that in the final reading of the document, a subtle misdescription of the results was identified in the figures 8 and 9. Therefore, we made the appropriate changes (lines 493-494 and 516-518). All changes in the current document are highlighted in yellow. Also, we have provided a point-by-point commentary for the reviewer's comments. We hope our efforts to reach the conditions you have criticized would be enough to have the paper accepted for publication at The Journal Physiology.

Sincerely, José Rodrigo Pauli, PhD.

EDITOR COMMENTS

Reviewing Editor:

Thank you for choosing the Journal of Physiology for your research. You have addressed the reviewer comments in a satisfactory way. Congratulations on this important contribution to the growing literature on metabolic effects of time-restricted feeding.

We appreciate your opinions. Thanks for contributing to our work.

Senior Editor:

Thank you for addressing the comemnts raised in the earlier review. The reviewers and reviewing editor are of the opinion that the manuscript is acceptable for publication. Please note however to comply with The Journal of Physiology statistics policy the authors must amend the manuscript, including the Figures to show exact p values (not *, **, *** in the Figures). Thank you for considering The Journal of Physiology to publish your work.

We agreed and we changed all statistics according to the journal specifications and we are thankfull for your comment.

REFEREE COMMENTS

Referee #1:

The authors present an interesting investigation of the sexually dimorphic responses to time restricted feeding in mice. Although the content is largely descriptive the recent revisions have enhanced the quality and the impact of the study making it a valuable contribution to the growing body of literature in this area.

We are thankful for this commentary and appreciate the consideration.

Referee #3:

Thank you for submitting your manuscript to The Journal of Physiology. The manuscript was selected for ethics review to confirm appropriate use of anaesthetic agents. The authors have provided details of the agents used, the doses used and the route of administration. The ketamine/xylazine mix is standard and appropriate. The authors did not perform surgical procedures. Rather, the mice were immediately euthanised by decapitation. The text is adequate as is. Thank you.

Line 136: Please add that the study conformed to the principles of The Journal of Physiology.

We are thankful for this commentary and all requests from the referee were attended (lines 133-134).

Dear Dr Pauli,

Re: JP-RP-2025-287681R2 "Time-Restricted Feeding Combined with Exercise Improves Hepatic and Glycemic Metabolism in Obese Mice: A Sex-Dependent Study." by Gabriel Calheiros Antunes, Camila Venturini Ayres Cunha, Leandro Kansuke Oharomari, Renan Fudoli Lins Vieira, Maura Fanti, Thaiane da Silva Rios, Luciana Renata Conceição de Mattis, Ana Paula Azevêdo Macêdo, Adelino Sanchez Ramos da Silva Silva, Eduardo R Ropelle, Dennys Esper Cintra, and Jose Rodrigo Pauli

Thank you for submitting your manuscript to The Journal of Physiology. It has been assessed by a Reviewing Editor and by 0 expert referees and we are pleased to tell you that it is acceptable for publication following satisfactory revision.

REVISION CHECKLIST:

Please upload two versions of your manuscript text: one with all relevant changes highlighted and one clean version with no changes tracked. The manuscript file should include all tables and figure legends, but each figure/graph should be uploaded as separate, high-resolution files. The journal is now integrated with Wiley's Image Checking service. For further details, see: <https://www.wiley.com/en-us/network/publishing/research-publishing/trending-stories/upholding-image-integrity-wileys->

image-screening-service

We look forward to receiving your revised submission.

Yours sincerely,

Paul Greenhaff
Senior Editor
The Journal of Physiology

EDITOR COMMENTS

Senior Editor:

Thank you for making the changes to the figures to show exact p values. However, the "Statistical analysis" section of the Methods still states "Data are presented as mean {plus minus} standard deviation (SD) and statistical significance was determined as $p < 0.05$ (* $p < 0.05$; ** $p < 0.01$; *** $p < 0.001$; **** $p < 0.0001$).

Thank you for the revised version. There is a further change required to the "Statistical analysis" section of the Methods to ensure the manuscript is acceptable for publication which the authors missed when revising the manuscript.

END OF COMMENTS

July 15, 2025

Dear Kim E. Barrett,

Editor-in-chief of The Journal of Physiology

Firstly, we would like to thank you for the opportunity to revise our paper. We have read the reviewer's comments and judged the suggestions very appropriately. We are confident that the edits within this revision have not only satisfied the reviewer's concerns but have significantly improved the manuscript. We hope that our efforts are evident from the responses below and the revised manuscript. All changes in the current document are highlighted in yellow. Also, we have provided a point-by-point commentary for the reviewer's comments. We hope our efforts to reach the conditions you have criticized would be enough to have the paper accepted for publication at The Journal Physiology. Sincerely, José Rodrigo Pauli, PhD.

EDITOR COMMENTS

Senior Editor:

Thank you for making the changes to the figures to show exact p values. However, the "Statistical analysis" section of the Methods still states "Data are presented as mean {plus minus} standard deviation (SD) and statistical significance was determined as $p < 0.05$ (* $p < 0.05$; ** $p < 0.01$; *** $p < 0.001$; **** $p < 0.0001$).

We agreed and we changed all statistical analysis in the Methods section.

Dear Dr Pauli,

Re: JP-RP-2025-287681R3 "Time-Restricted Feeding Combined with Exercise Improves Hepatic and Glycemic Metabolism in Obese Mice: A Sex-Dependent Study." by Gabriel Calheiros Antunes, Camila Venturini Ayres Cunha, Leandro Kansuke Oharomari, Renan Fudoli Lins Vieira, Maura Fanti, Thaiane da Silva Rios, Luciana Renata Conceição de Mattis, Ana Paula Azevêdo Macêdo, Adelino Sanchez Ramos da Silva Silva, Eduardo R Ropelle, Dennys Esper Cintra, and Jose Rodrigo Pauli

We are pleased to tell you that your paper has been accepted for publication in The Journal of Physiology.

Yours sincerely,

Paul Greenhaff
Senior Editor
The Journal of Physiology

If you would like to receive our 'Research Roundup', a monthly newsletter highlighting the cutting-edge research published in The Physiological Society's family of journals (The Journal of Physiology, Experimental Physiology, Physiological Reports, The Journal of Nutritional Physiology and The Journal of Precision Medicine: Health and Disease), please click this link, fill in your name and email address and select 'Research Roundup':
<https://www.physoc.org/journals-and-media/membernews>

- You can help your research get the attention it deserves! Check out Wiley's free Promotion Guide for best-practice recommendations for promoting your work at: www.wileyauthors.com/eeo/guide. You can learn more about Wiley Editing Services which offers professional video, design, and writing services to create shareable video abstracts, infographics, conference posters, lay summaries, and research news stories for your research at: www.wileyauthors.com/eeo/promotion.

EDITOR COMMENTS

Senior Editor:

Thank you for making the final requested changes to the manuscript. It is now acceptable for publication. Thank you for

considering to publish your work in The Journal of Physiology.